# Exploring the Changes in Risk Perceptions and Adaptation Behaviors Based on Various Socioeconomic Characteristics Before and After Earthquake Disasters – A Case Study in Taiwan

Tzu-Ling Chen[1], Tzu-Yuan Chao[2], Hao-Tung Cheng[3]

[1] Department of Urban Development, University of Taipei, Taipei City, 111, Taiwan
[2] Department of Urban Planning, National Cheng-Kung University, Tainan City, 701, Taiwan
[3] Department of Urban Planning, National Cheng-Kung University, Tainan City, 701, Taiwan

*Correspondence to*: Tzu-Ling Chen (skylight@mail2000.com.tw)

**Abstract.** Resilience, which has rapidly become an area of interest in multiple disciplines, is regarded as being key in disaster mitigation and adaptation. The objective indicator framework is a common way to evaluate resilience, but limited attention has been paid to measuring the risk perceptions and adaptation behaviors of individuals. In addition, due to limitations related to predicting potential earthquake events, past studies have placed more emphasis on predisaster discussions. Fortunately, this paper explores the changes in risk perceptions and adaptation behaviors in different socioeconomic groups through a

comparative analysis between pre- and postearthquake disasters, and one-way analysis of variance (ANOVA) with a post hoc test is applied to examine the changes in risk perceptions and adaptation behaviors. The results show that people tend to have greater risk perceptions of future earthquakes but were less willing to retrofit their houses after a serious disaster. Females show greater fear and worry accompanied by a higher willingness to retrofit their houses compared to males. In addition, people with a higher education level and a better occupation might be more willing than others to adopt adaptation behaviors.

For females and people with lower education levels, the results can serve as a reference to provide risk communication, risk education, and diverse disaster adaptation options. Although limitations exist, the results of comparative analysis between the predisaster and postdisaster conditions could serve as a reference for adequate strategies and government decisions on the prioritization of risk management policies.

## 1 Introduction

The Ring of Fire in East Asia has been regarded as the region most frequently hit by earthquake disasters because of the high rate of earthquakes that have previously occurred there compared to the global rate (USGS 2017). The call for disaster prevention and risk reduction has been made since the declaration of the International Decade for Natural Disaster Reduction in 1999 (UNISDR 1999). To mitigate dramatic losses, governments have invested a great amount of public resources to finance disaster management, and in particular, structural engineering measures are the major approaches taken to cope with

earthquake events. However, the risk of property damage and loss of life is possible wherever development is allowed in potential seismic areas because the occurrence of disasters may be at or below the design standard incorporated in building

codes and structural work areas (Kerr et al. 2003; Petak and Atkisson 1982; Sheaffer and Roland 1976). The disadvantage of the common reliance on structural engineering measurements has resulted in a new research focus on mediating the exposure to risk by selecting suitable adjustments. Recently, the Sendai Framework for Disaster Risk Reduction 2015-2030 has stipulated that the main priorities for disaster mitigation and adaptation are minimizing disaster risk and building resilience (UNISDR 2019).

It is necessary to minimize disaster risk and build resilience by self-evaluating the capabilities and capacities in responding to risk, that is, preparedness (Jones and Tanner 2017). Being prepared for a future disaster requires various components, such as sufficient personal character, social connections, and financial affordability (Baker and Cormier, 2015). People who are included in vulnerable minority groups and marginalized people might not be able to prepare in advance (Blake et al., 2017). Therefore, an increasing number of studies have emphasized measuring risk perceptions at the individual and household levels (Brown and Westaway 2011; Adger et al. 2009). The perception of disaster risk does not represent a direct function of the probability that threatening events will occur; rather, risk perception captures many other factors, such as attitude, cognition, the degree of danger comprehension, and vulnerability (Sjöberg 2000; Sjöberg 1996; Eagly and Chaiken 1993). Despite the substantial literature illustrating the origin (Barrows, 1923), concept (Sjöberg 2000; Sjöberg 1996), formation (Lindell et al., 2016; Whitney et al., 2004; Wu and Lindell, 2004; Lindell and Perry, 2000), and physical and social contexts of disaster risk perceptions (Blanchard-Boehm and Cook, 2004; Peacock et al., 2005; Peacock, 2003), less attention has been paid to systematically examining changes in risk perceptions.

In fact, disaster experiences might facilitate or constrain preparedness (Becker et al., 2017; Ejeta et al., 2015; Lindell and Perry, 2011; Bostrom, 2008), and such effects might be biased across disasters, cultures or regions. A disaster resulting in limited impacts or the assumption that a future disaster will not occur might encourage people to not prepare for future disasters (Paton et al., 2014; Barron and Leider, 2010). Alternatively, people might take any adaptation approaches based upon damage or losses, physical injury, emotional injury and so on (Perry and Lindell, 2008; Nguyen et al., 2006; Heller et al., 2005). The physical damage or losses (Solberg et al., 2010) and psychological fear or anxiety (Rüstemli and Karanci, 1999) resulting from disaster experiences could motivate adaptation behaviors. However, socioeconomic characteristics such as income, age, and gender might encourage or discourage individuals from taking adaptive actions (Bankoff 2006; Wisner et al. 2004). For example, if people cannot act adequately to mitigate such anxiety, they might take no actions at all (Paton and McClure, 2013). Due to limited knowledge and resources, people tend not to respond to common disasters and tend to have personal preferences for disasters, such as denying disasters, denying disaster probability, and having certain beliefs about the government and public infrastructure. Therefore, examining risk perceptions and adaptation behaviors based on various socioeconomic characteristics could provide important information for disaster management.

In summary, the threats in a given area posed by future earthquakes with a magnitude larger than that experienced in the past create uncertainty in regard to the ability to mitigate impacts to acceptable levels using only engineering or construction measures. Humans have the capacity to respond to the environment to reduce risk by learning from past experience, and changes in attitudes and behaviors are very helpful in responding to earthquake disasters (Gifford 2014). Theoretically, a more

accurate measurement and tracking of the interactions of socioeconomic characteristics that collectively affect responses to disasters might help support the right activities and target the right people in disaster management (Oddsdottir et al. 2013; Adger 2000). Past studies have placed more emphasis on predisaster conditions to explore the interaction effects of various socioeconomic characteristics on individuals' decisions (Levine 2014). Examining predisaster and postdisaster conditions could reveal the impact of extreme events and how people's perceptions of such events and their willingness to take potential adaptation approaches might change. Therefore, this study contributes by exploring how earthquake disasters influence the risk perceptions and adaptation behaviors of residents in Taiwan and further categorizes them according to socioeconomic characteristics. The sample is of particular interest because it contains pre- and postdisaster information on residents who were directly affected by the Meinong earthquake (participants completed surveys approximately 1 year before and 3 months after the earthquake), allowing a more robust analysis of the effects of natural disasters on subjective resilience compared to previous research. Based on past studies, the interactions of socioeconomic characteristics can collectively affect responses to disasters. This study discusses such responses based on various socioeconomic characteristics to explore how such characteristics affect pre- and postearthquake risk perceptions and adaptation behaviors. In addition to the introduction, this paper is organized as follows. Section 2 provides a brief description of the research design, including the study area, the data collection, the measures for subjective resilience, and the methods. Section 3 presents the comparative analysis between pre- and postdisaster surveys based on the results of one-way analysis of variance (ANOVA). Section 4 presents the comparative analysis between our findings and those of past studies. The final section offers some conclusions.

## 2 Data and Methodology

### 2.1 Study area

The study area of Taiwan is located along the Philippine Sea Plate and the Eurasian Plate, and the orogenic belt of central-southern Taiwan has undergone intensive crustal deformation. It is exposed to earthquake events, as most active faults were confirmed after the city had already been built on them. An active fault called the Houchiali Fault trends north to south across the study area (Lin et al., 2000; Chen and Liu, 2000). Although the existing Houchiali Fault has recently been identified as a Late Pleistocene active fault, an intensified and densely built environment has developed right on and close to the fault line (see Fig. 1b). In addition, there is increasing population growth in the study area, and in particular, some areas along the fault line have a relatively densely clustered population (see Fig. 1c, 1d). In addition, the soft soil might amplify surface ground motion. In 2015, the Meinong earthquake, a local magnitude 6.6 earthquake, struck southern Taiwan, having a devastating impact and resulting in 117 deaths; additionally, numerous buildings were reported to have collapsed (National Applied Research Laboratories 2018; Tsai et al. 2017). Previously unspecified regulations resulted in a number of 5-story buildings without earthquake safety. In the study area of Yongkang, 744 buildings were reported as having been damaged, and in particular, one building fully collapsed, resulting in 115 deaths (see Fig. 1a). According to the Central Weather Bureau (Huang et al. 2009), a large magnitude earthquake occurs once every thirty years in southern Taiwan. A low willingness to make

repairs was found, even though the government encourages inhabitants with this low willingness to retrofit buildings through subsidies and tax relief. The Expediting the Reconstruction of Urban Unsafe and Old Buildings statute was quickly promulgated on May 10, 2017.

## 2.2 Data collection

There are thirty-nine townships within the study area. A total of 429 individuals completed the predisaster survey, which was conducted between October and December 2014. The postearthquake follow-up survey was conducted in May 2016 (3 months after the Meinong earthquake), and trained interviewers conducted the survey over the phone, asking the same questions as those in the predisaster survey. All survey sampling methods relied on voluntary response sampling. The predisaster survey was a street survey, while the postdisaster survey was a telephone survey based on phone number databases within the study area and conducted by the survey research center of a domestic academic institution. The respondents were reminded of some particular information regarding the most recent earthquake, the geographic location of the nearest fault line, the impact of the disaster event, the frequency of earthquakes in the study area, etc. Additionally, the scale of earthquake magnitude is defined as over 6.0. The content of the survey questions contained five parts: behavioral intentions to adopt residential seismic strengthening, risk perceptions, sensitivity to earthquakes, trust in the government and responsibility attribution. All parts contain at least three items. The main goal of our study is to explore the trajectory of risk perceptions and adaptation behaviors before and after the Meinong earthquake. The same questionnaire allows us to examine these issues with the same earthquake risk area 1 year before and 3 months after this disaster.

## 2.3 Measures for risk perceptions and adaptation behaviors

Perceived risk is not necessarily equivalent to the probability of occurrence of a disaster. Rather, it summarizes many other factors. Increasing research focuses on the risk perceptions of earthquake disasters, and such perceptions might vary. Previous studies have shown that terror often accompanies changes in the physical environment, the loss of human lives and the destruction of property. Therefore, among earthquake-related stressors, we were concerned with individuals' perceptions of the probability of an earthquake disaster occurring within ten years and the impacts they expected from such a disaster, including fear of earthquakes and worries over buildings collapsing.

Although prior disaster experiences and observation of the natural environment might form disaster perceptions, various socioeconomic characteristics might further affect such perceptions. Adaptation behavior is a way for individuals to adapt their living environment to new events that may occur and impact the existing system. People who have faith in adaptation behaviors might take whatever approaches they have, while others might take no such approaches. Therefore, in the adaptation behavior section, we were concerned with the ways in which people respond to earthquake disasters. To survive earthquakes, seismic

restraints might play important roles during such disasters. Hence, there are two items regarding house retrofitting, including the willingness to retrofit houses and house retrofitting after professional assessment.

There are five items in the survey to explore both risk perceptions and adaptation behaviors. Risk perceptions are measured by three items on the expected impacts of earthquakes, and adaptation behaviors are measured by two items on the willingness to support policies. The measurement, shown in Table 1, combines 7-point Likert-scale items and Yes/No questions (see Table 1). A transformation process is conducted to solve the problems posed by scales with different measurement systems.

## 2.4 Methods: one-way analysis of variance

One-way ANOVA is an extension of the independent samples t-test that can be used to compare any number of groups (Bewick et al. 2004; Whitely and Ball 2002). The core value of one-way ANOVA lies in the ability to examine means that are significantly different from each other between groups. One-way ANOVA is calculated as follows:

$$\frac{\sum_{i=1}^{n}(x_i - \bar{x})^2}{n-1} \tag{1}$$

where the variance comes from a set of n values $(x_1, x_2, \dots, x_n)$ and the degrees of freedom is n-1.

In one-way ANOVA, the F statistic test is used and represented equally among groups. A significant F statistic test result indicates a significant difference between groups, and the P-value of 0.05 is the common threshold. First, Levene's test is applied to examine the null hypothesis that the variance is equal across groups. A result of Levene's test lower than 0.05 indicates that it is necessary to apply Welch's test because there is no equal variance between groups. On the other hand, if the result of Levene's test is greater than 0.05, then we can depend on the ANOVA results. Overall, a significant F statistic in both Welch's test and ANOVA indicates that at least two groups are different, but it does not identify which groups are different from the others. However, a P-value lower than 0.05 indicates significance or the probability of a type II error, which is the possibility of incorrectly rejecting the null hypothesis or wrongly concluding a difference between groups. Therefore, a post hoc test and multicomparison analysis testing are necessary to avoid type II errors and to further examine the differences between levels. Due to the assumption of homogeneity of variance, we then apply the Games-Howell test and Benjamini-Hochberg procedure.

Quantitative data analysis was conducted using the Statistical Package for Social Scientists (SPSS) software. Each response to the items in the questionnaire survey was rated on a scale ranging from 1 to 7, with 1 as the highest level of vulnerability (or lowest level of resilience) and 7 as the lowest level of vulnerability (highest level of resilience).

## 3 Results

The number of respondents was similar across genders, which is consistent with the gender ratio in the study area. Regarding age, most respondents in the pre- and postearthquake surveys were between 16 and 60 years old and thus had the knowledge and capacity to develop their self-perceptions and adaptation behaviors. Regarding education, most residents in the study area were university graduates. Because the survey was based on voluntary response sampling, the results showed that there might be inconsistencies in the education category because most respondents graduated from high school. In terms of

occupation, the official statistics exclude students and home makers from the labor force. In Taiwan, we have only the national statistics of the industry and service census[1]. Therefore, the overall occupation ratio in the study area can be divided into two categories: employment and unemployment. In Taiwanese culture, owning one's house is preferred over renting. Indeed, the survey shows that less than 20% of the respondents rent their homes (see Table 2). In general, people became highly aware of earthquakes immediately after the Meinong earthquake, but people were unwilling to retrofit their houses. In the following

sections, the study attempts to compare risk perceptions and adaptation behaviors pre- and postdisaster based on socioeconomic characteristics such as gender, age, education, occupation, and house ownership.

### 3.1 Gender

        In the preearthquake survey, males showed more worries than females regarding building collapsing (P-value = 0.008 < 0.05), while the results for the other items were not statistically significant. In the postearthquake survey, the probability of an

earthquake disaster occurring within the next ten years (P-value = 0.049 < 0.05), fear of earthquakes (P-value = 0.000000 <0.05), and the willingness to retrofit houses (P-value = 0.002 < 0.05) were statistically significant, indicating variations between the gender categories. The results show that the Meinong earthquake not only increased awareness of earthquakes but also increased the risk perceptions of females (probability of an earthquake disaster: 4.74 (females) > 4.51 (males); fear of earthquakes: 5.64 (females) > 4.75 (males)). Both males and females were less willing to retrofit their houses after this serious

earthquake. In summary, although the coefficient of risk perception among males is higher than that among females in the preearthquake survey, the coefficient among males becomes lower than that among females in the postearthquake survey. In addition, there is significant variation between gender categories after the Meinong earthquake, and females show higher risk perceptions and a higher willingness to retrofit their houses than males (see Table 3).

### 3.2 Age

According to the F-test, the result for worries over buildings collapsing is statistically significant (P-value = 0.045 < 0.05) in the postearthquake survey (see Table 4). To examine whether there are variations, this study applied the Hochberg test to explore such variations. However, the results of the Hochberg test show that there are no statistically significant differences between age groups. Therefore, the overall results show that there are no significant variations among age categories in both

---

[1] https://eng.stat.gov.tw/np.asp?CtNode=1548

the pre- and postearthquake surveys. Because there are no variations among age groups, we use the mean value to compare
the changes between the pre- and postearthquake surveys. In terms of risk perceptions, people tended to become more aware
of earthquakes (probability of an earthquake disaster: 4.04 (pre) < 4.55 (post); fear of earthquakes: 4.91(pre) < 5.02 (post); and
worries over buildings collapsing: 4.61 (pre) = 4.61 (post)). Regarding adaptation behaviors, people tended to become less
willing to retrofit their houses. Therefore, the overall results show that there are no significant variations among age categories
both the pre- and postearthquake surveys. It seems that age does not necessarily affect risk perceptions or adaptation behaviors.

## 3.3 Education

Again, in the preearthquake survey, there are no significant variations among education categories, indicating that
different educational level groups show a similar awareness of the probability of earthquakes and a similar willingness to
retrofit their houses. In contrast, the results regarding the probability of an earthquake disaster occurring within the ten years
(P-value = 0.001 < 0.05), worries over buildings collapsing (P-value = 0.046 < 0.05), and willingness to retrofit houses after
assessment (P-value = 0.005 < 0.05) are statistically significant, indicating significant differences among educational level
categories (see Table 5). This paper further applies post hoc analysis to compare the differences between categories. The results
show that different educational level categories do indeed have different levels of awareness of the probability of earthquakes
and preferences for house retrofitting. For example, one variation (-0.579) shows that people who graduated from elementary
or junior high school might have less awareness than people who graduated from university or graduate school. Meanwhile,
another variation (-0.42) shows that people who graduated from elementary or junior high school might be less willing to
retrofit their houses (see Table 6). Overall, people tended to become more aware of earthquakes after the Meinong earthquake
and less willing to retrofit their houses. Although there are no significant results showing that education matters for risk
perceptions and adaptation behaviors, after the Meinong earthquakes, those with a higher educational level seemed to become
more aware of the probability of earthquakes and willing to retrofit their houses.

## 3.4 Occupation

The results show that occupation matters for both risk perceptions and adaptation behaviors in both the pre- and
postearthquake surveys. According to the F statistic test, the results for the items on fear of earthquakes (P-value = 0.004 <
0.05) and worries over buildings collapsing (P-value = 0.005 < 0.05) in the preearthquake survey (see Table 7) are statistically
significant. The results of the Hochberg test show that home makers have higher risk perceptions than white-collar workers,
blue-collar workers, and students (see Table 8). In the postearthquake survey, the results for the probability of an earthquake
disaster occurring within the next ten years (P-value = 0.016 < 0.05), fear of earthquakes (P-value = 0.000142 <0.05), worries
over buildings collapsing (P-value = 0.018 < 0.05), willingness to retrofit houses (P-value = 0.008 < 0.05), and willingness to
retrofit houses after assessment (P-value = 0.036 < 0.05) are all statistically significant, indicating significant differences
between occupation categories (see Table 7). The results of the post hoc test show that home makers have the highest awareness
of the risk of earthquakes among all occupation categories. In terms of house retrofitting, there are significant variations

between white-collar and blue-collar workers. In summary, after the Meinong earthquake, regardless of occupation, people tended to become more aware of earthquakes but less willing to retrofit their houses. In addition, home makers are much more aware of earthquake risk than those holding other occupations in both the pre- and postearthquake surveys. Due to their economic status, white-collar workers tended to be more willing to retrofit their houses after the Meinong earthquake compared to blue-collar workers.

### 3.5 House ownership

Regarding house ownership, most categories show no statistically significant variations in the pre- and postearthquake surveys (see Table 9). In the postdisaster survey, the P-value (0.009 < 0.05) for the willingness to retrofit houses indicates that at least two house ownership groups have significantly different preferences. This paper further applies the post hoc test examine the different preferences for house retrofitting (see Table 10). The results show that the family-owned group has a higher willingness to retrofit houses than the self-owned group in the postearthquake survey. Overall, regardless of house ownership category, people tended to become more aware of earthquakes and less willing to retrofit their houses in the postearthquake survey. Although there are no particular variations in risk perceptions among the house ownership categories, people who owned their house still show a higher willingness to retrofit their houses compared to those who rented.

### 4 Discussion

According to the results, after the Meinong earthquake, people tended to have greater risk perceptions regarding future earthquakes but were less willing to retrofit their houses. The findings show that people might become less willing to prepare, which is quite similar to the result of a survey conducted after the 2011 Christchurch earthquake (Statistics New Zealand, 2012; Paton and Johnston, 2008). In fact, the relationship between disaster experience and preparedness has been regarded as a key issue based on the recommendations of the Sendai Framework (United Nations, 2015). According to past studies, it is difficult for people to imagine any consequences if they lack earthquake experience (Paton and McClure, 2013). However, the study finds that the levels of disaster preparedness become low after serious disasters. Therefore, disaster experience might not necessarily increase people's willingness to prepare. On the other hand, socioeconomic characteristics might still affect the decision-making process with regard to adopting adaptation behaviors.

In terms of gender, females show greater fear and worries regarding future earthquake disasters than males, while they have a similar willingness to retrofit their houses (see Fig. 3). According to past studies, the responses of women might be more internal and backstage, whereas those of men might be more external and front stage (Enarson 2001; Always et al. 1998; Fordham 1998). The economic status and family role of women might forbid possible adaptive choices compared to men (Tobin-Gurley and Enarson 2013). Men, in contrast, are more risk tolerant than women (Finucane et al. 2000). Although gender inequality prevails in different ways around the world, women's safety concerns for their family have been well documented in both environmental protection movements and neighborhood emergency preparedness campaigns (Litt et al. 2012; Luft

2008; Erikson 1994; Turner et al. 1986). Therefore, it is necessary to provide more diverse options for house retrofitting for families to increase their potential willingness to improve the anti-seismic resilience of their houses.

Regarding education, people tend to become aware of earthquake risk after a serious disaster event, and there are no
significant variations between educational level categories. Although there is a significant decrement in the result for house retrofitting, people who have a university-level education might be more willing to retrofit their houses (see Fig. 4). There are similarities in occupation; people who are white-collar workers are still much more willing to retrofit their houses than blue-collar workers, home makers, and students. In addition, home makers have higher risk perceptions than those belonging to the other occupation categories. Available resources might be the key factor affecting whether people prepare for and respond to
disasters. Social stratification plays a role in perceiving and reacting to risk, including people's understanding of disaster information, the sources announcing disaster information, and potential options to respond (Fothergill and Peek 2004).

Gender, age, and class alone do not make people vulnerable, while the interactions between factors might result in an increase in vulnerability. Overall, social characteristics do indeed affect decisions regarding disaster awareness and adaptation behaviors. In addition, disaster experience does indeed facilitate local awareness but constrains preparedness in regard to
Taiwan's earthquake experience. Among gender, education, and occupation, each category shows a similar tendency of increased risk awareness of risk but decreased willingness to retrofit houses. However, over time, risk awareness might fade away. Therefore, risk communication, risk education, and diverse mitigation options are required as soon as possible after serious earthquakes to help people be ready for future events.

## 5 Conclusions

Our comparative analysis of predisaster and postdisaster surveys based on various socioeconomic characteristics contributes to the significant and meaningful results of this study. The study found that the responses to earthquake disasters being varied between male and female which consistent with past findings. Although disaster experience does indeed play an important role in helping people become aware of earthquake risk, disaster experience is not necessary increase people's willingness on house retrofit. In addition, people with higher education and occupation represents who might have more
available resources and therefore they might become more willing to prepare for and respond to disasters. Although gender, age, and class alone do not make people vulnerable, the interaction between various socioeconomic characteristics might result in an increase in vulnerability to disasters.

This study tends to explore the changes in risk perceptions and adaptation behaviors based on various socioeconomic characteristics before and after earthquake disasters. However, there are multiple limitations faced in this study. There are two
surveys (October to December 2014, and May 2016) conducted in the study area. The predisaster survey was a street survey, while the postdisaster survey was a telephone survey based on phone number databases within the study area. Although the questions were the same in the two surveys, the interviewees in the pre- and postearthquake surveys were different. In addition, Meinong earthquake was a magnitude 6.6 earthquake caused 744 buildings reported as having been damaged, and in particular, one building fully collapsed, resulting in 115 deaths. It was a devastating earthquake but only caused one building fully

collapsed. Such disaster experience might not necessary increase the awareness of buildings anti-seismic effect. The results might not be applicable to any other disaster events, only earthquakes.

To sum up, the results can provide a general tendency regarding changes in risk perceptions and adaptation behaviors pre- and postdisaster events and the variations between different socioeconomic characteristics based upon Taiwanese disaster experience. The findings can serve as a reference to formulate risk communication strategies and for governments to make

decisions on the prioritization of risk management policies. However, there are potential topics that could be extended in future studies, such as the correlation between socioeconomic characteristics, the causes and effects of risk perceptions on adaptation behaviors, and others.

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

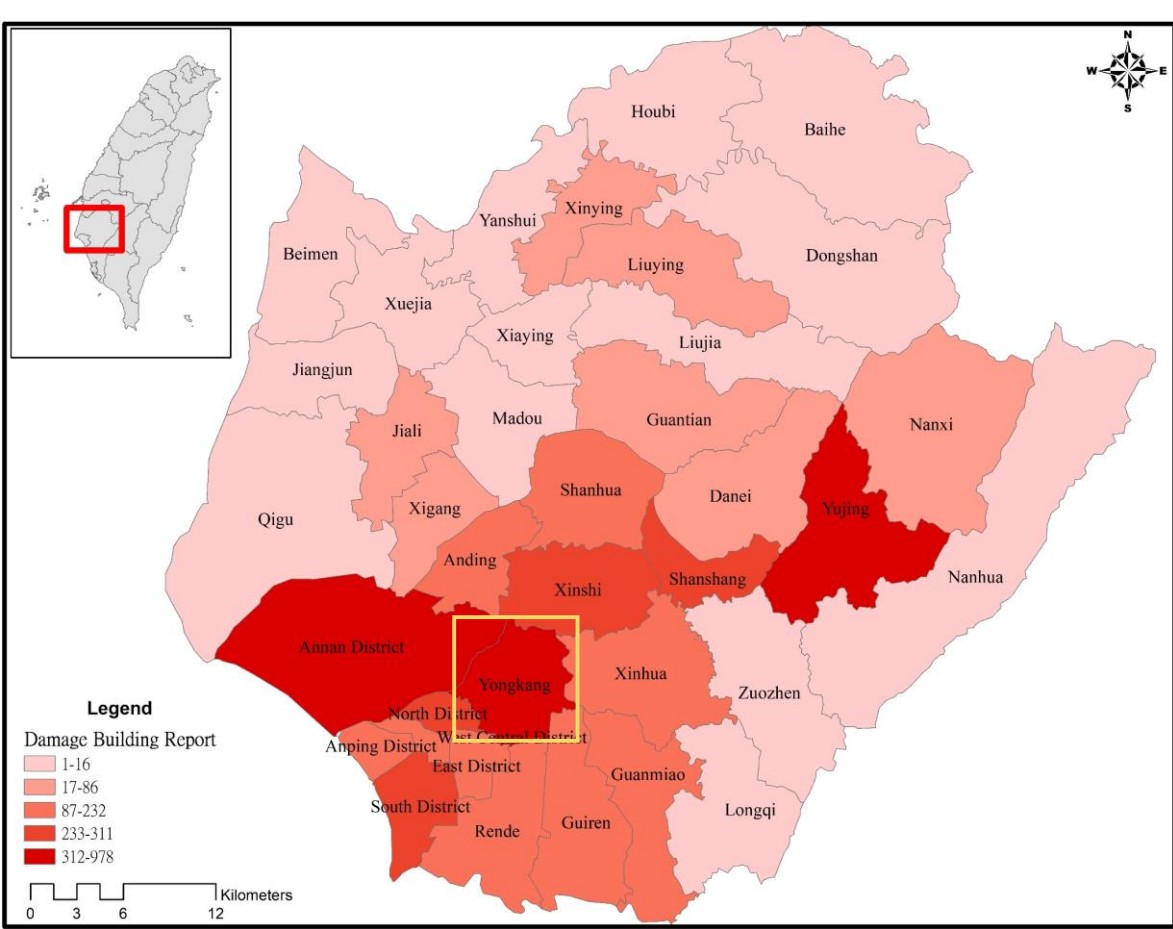

(a) Damaged buildings in the Meinong earthquake in Tainan City

**Fig. 1.** Study area


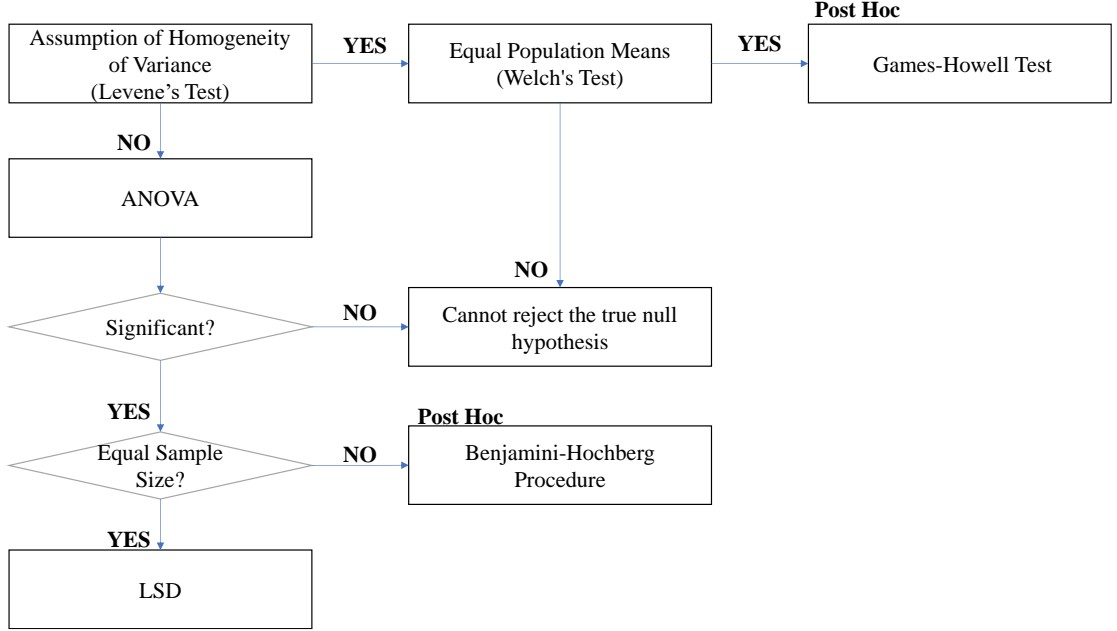

**Fig.** 2. Overall process of one-way analysis of variance

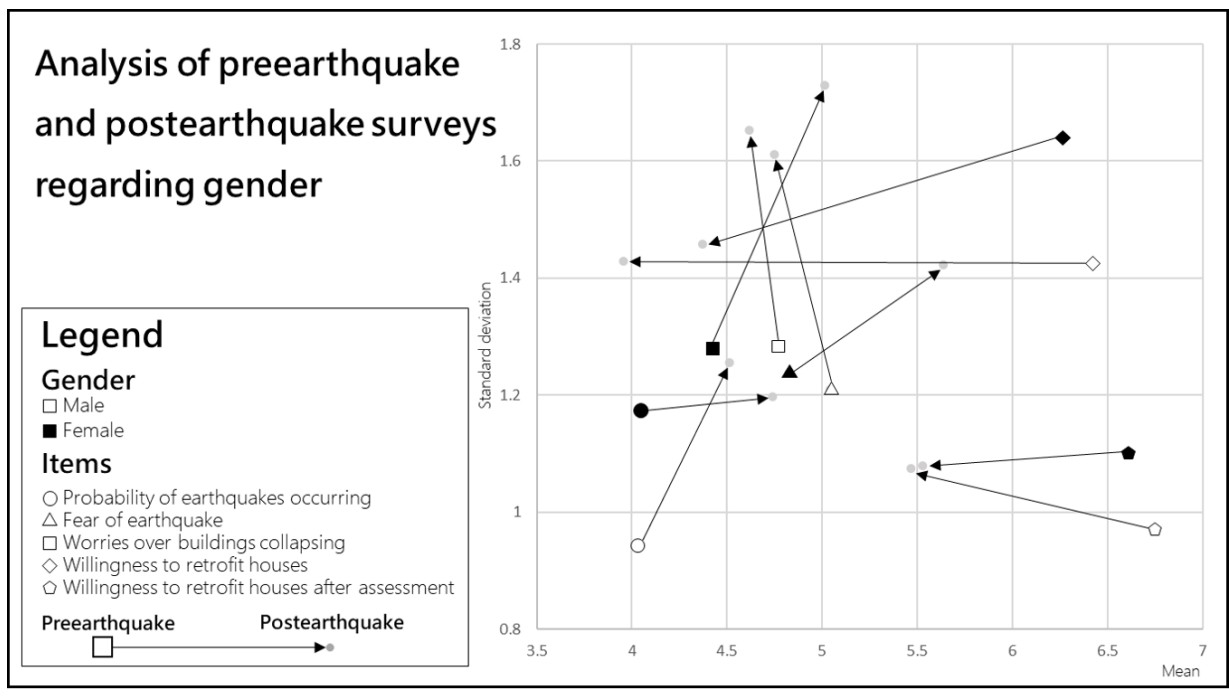

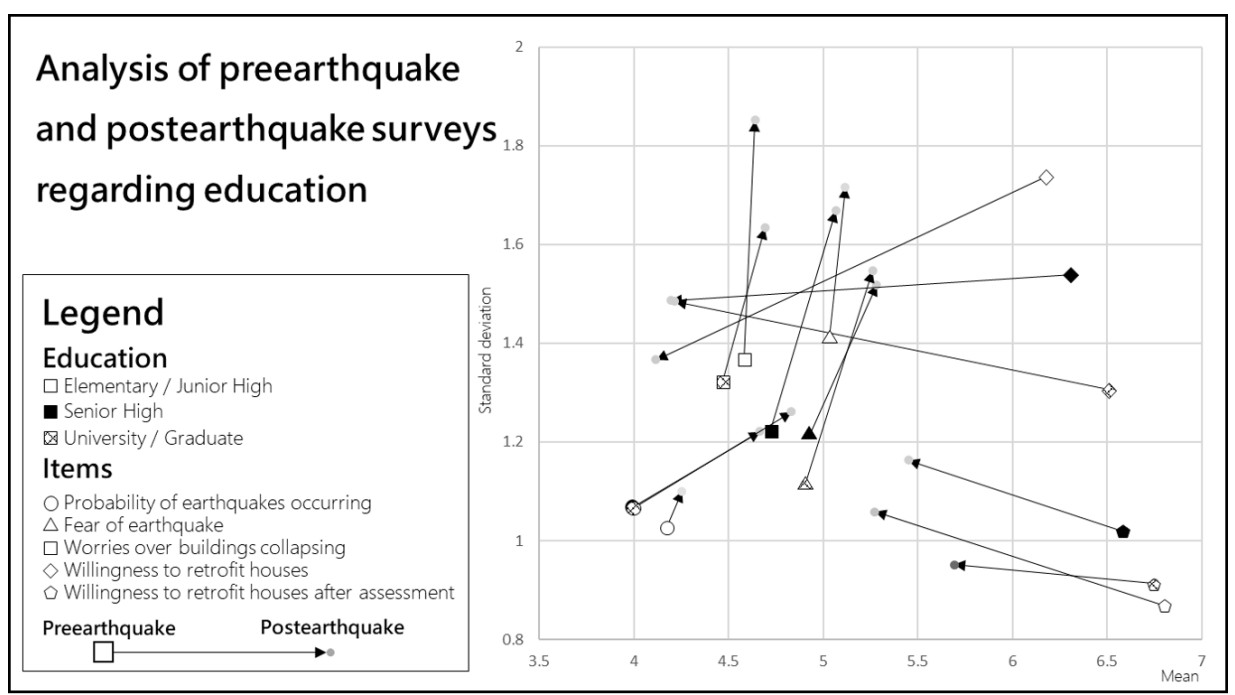

**Fig.** 4. Comparative analysis of the pre- and postearthquake surveys regarding education

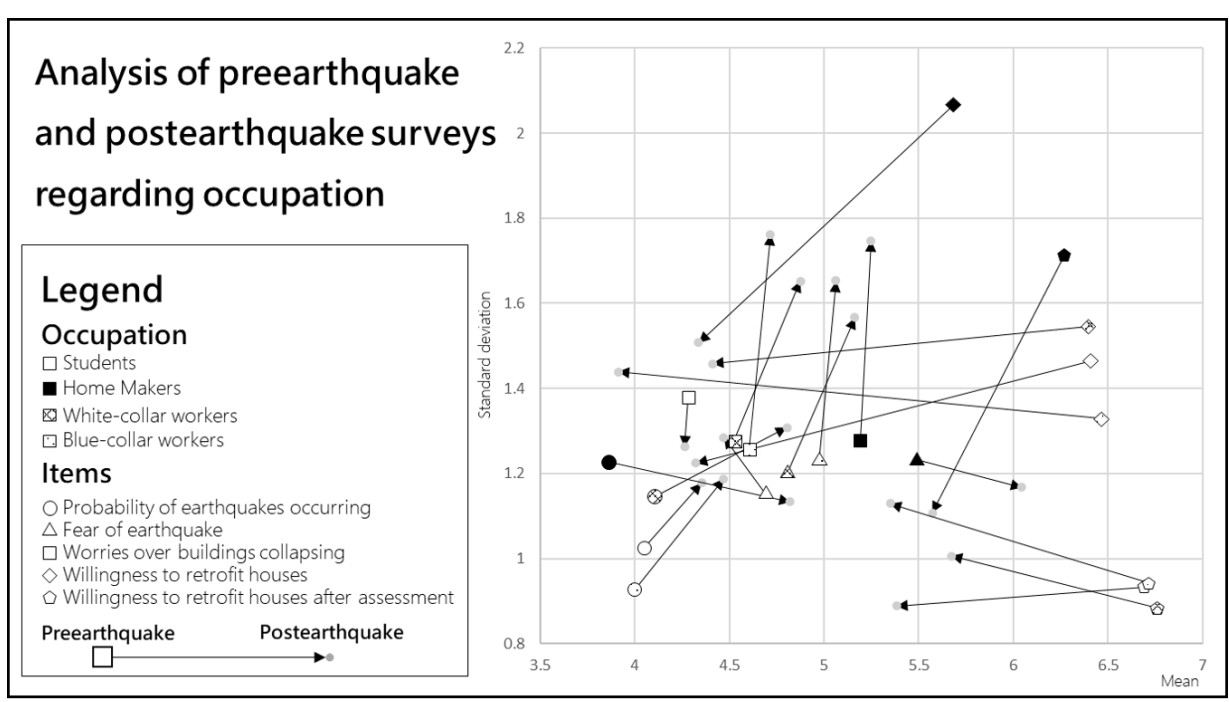

**Fig.** 5. Comparative analysis of the pre- and postearthquake surveys regarding occupation

**Table 1** Measurement of the questionnaires.

| Aspects | Items | predisaster | postdisaster |
|---------|-------|-------------|--------------|
| Risk perceptions | Probability of an earthquake disaster occurring within the next ten years | 7-point | 7-point |
| | Fear of earthquakes | 7-point | 7-point |
| | Worries over buildings collapsing | 7-point | 7-point |
| Adaptation behaviors | Willingness to retrofit houses | Yes/No | 7-point |
| | Willingness to retrofit houses after assessment | Yes/No | 7-point |

Completely disagree = 1 to completely agree =7

**Table 2** Sample characteristics in the pre- and postearthquake surveys.

| Characteristics | Pre- | Post- | Study area | Characteristics | Pre- | Post- | Study area |
|-----------------|------|-------|-----------|-----------------|------|-------|-----------|
| **Gender** | | | | **Occupation*** | | | |
| Male | 53.38% | 44.89% | 49.27% | Students | 9.09% | 7.23% | 38.53% |
| Female | 46.42% | 55.11% | 50.73% | Home Makers | 10.96% | 18.94% | |
| **Age** | | | | White-collar Workers | 37.76% | 32.55% | 59.08% |
| < 15 yr. | 7.46% | 1.70% | 13.97% | Blue-collar Workers | 41.96% | 41.28% | |
| 15-40 yr. | 38.23% | 28.30% | 37.96% | **House Ownership*** | | | |
| 40-60 yr. | 37.53% | 51.91% | 32.16% | Self-owned | 48.95% | 63.62% | 85.93% |
| > 60 yr. | 16.78% | 18.09% | 15.91% | Family-owned | 32.17% | 32.34% | 3.20% |
| **Education** | | | | Renting | 18.65% | 4.04% | 7.82% |
| Elementary/Junior High | 21.68% | 21.91% | 21.63% | | | | |
| High School | 47.32% | 41.49% | 30.54% | | | | |
| University/Graduate | 31.00% | 36.60% | 46.96% | | | | |

Note 1: The values without official statistics are replaced by data from the Tainan Municipality.
Note 2: The share of illiterate individuals in the study area is 0.87%.
Note 3: The official statistics for occupation are categorized into employment and unemployment, and the unemployment percentage is 2.39%. In addition, neither students nor home makers are included in the labor force.

Note 4: The official statistics for house ownership include self-owned, family-owned, renting, and others, and the percentages are 85.93%, 3.20%, 7.82%, and 3.05%, respectively.

**Table 3** P-values and means for gender.

| Items | Preearthquake | | | | | Postearthquake | | | | |
|---|---|---|---|---|---|---|---|---|---|---|
| | Male | Female | DF | T | P-value | Male | Female | DF | T | P-value |
| Probability of an earthquake disaster occurring within the next ten years | 4.03 | 4.05 | 415 | -0.211 | 0.836 | 4.51 | 4.74 | 468 | -1.988 | 0.049* |
| Fear of earthquakes | 5.04 | 4.85 | 415 | 1.643 | 0.101 | 4.75 | 5.64 | 468 | -6.342 | 0.000[1]*** |
| Worries over buildings collapsing | 4.77 | 4.44 | 415 | 2.644 | 0.008** | 4.62 | 5.02 | 468 | -2.539 | 0.011* |
| Willingness to retrofit houses | 6.42 | 6.23 | 415 | 1254 | 0.218 | 3.96 | 4.37 | 468 | -3.085 | 0.002** |
| Willingness to retrofit houses after assessment | 6.75 | 6.58 | 415 | 1.485 | 0.123 | 5.46 | 5.53 | 468 | -0.646 | 0.519 |

[1]0.000000

* $p < 0.05$; ** $p < 0.01$; ***$p < 0.001$; 0.000***

<p style="text-align:center">Table 4 P-values and means for age.</p>

| Items | Preearthquake | | | | | | |
| --- | --- | --- | --- | --- | --- | --- | --- |
| | < 15 yr. | 16-40 yr. | 41-60 yr. | > 61 yr. | DF | F | P-value |
| Probability of an earthquake disaster occurring within the next ten years | 4.00 | 4.02 | 4.00 | 4.15 | 428 | 0.372 | 0.773 |
| Fear of earthquakes | 4.68 | 4.88 | 5.01 | 5.07 | 428 | 1.135 | 0.334 |
| Worries over buildings collapsing | 4.66 | 4.67 | 4.57 | 4.54 | 428 | 0.248 | 0.863 |
| Willingness to retrofit houses | 6.41 | 6.44 | 6.30 | 6.21 | 428 | 0.463 | 0.708 |
| Willingness to retrofit houses after assessment | 6.50 | 6.61 | 6.72 | 6.83 | 428 | 1.121 | 0.340 |
| Items | Postearthquake | | | | | | |
| | < 15 yr. | 16-40 yr. | 41-60 yr. | > 61 yr. | DF | F | P-value |
| Probability of an earthquake disaster occurring within the next ten years | 4.38 | 4.69 | 4.67 | 4.45 | 466 | 0.935 | 0.424 |
| Fear of earthquakes | 4.50 | 5.17 | 5.39 | 5.02 | 466 | 1.955 | 0.120 |
| Worries over buildings collapsing | 4.13 | 4.72 | 5.04 | 4.54 | 466 | 2.701 | 0.045* |
| Willingness to retrofit houses | 4.88 | 4.26 | 4.20 | 3.98 | 466 | 1.285 | 0.279 |
| Willingness to retrofit houses after assessment | 5.50 | 5.53 | 5.52 | 5.39 | 466 | 0.365 | 0.779 |

$* p < 0.05$; $** p < 0.01$; $***p < 0.001$.


**Table 5** P-values and means for education.

| Items | Preearthquake | | | | | |
| --- | --- | --- | --- | --- | --- | --- |
| | Elementary/ Junior High | Senior High | University/ Graduate | DF | F | P-value |
| Probability of an earthquake disaster occurring within the next ten years | 4.17 | 4.00 | 4.00 | 428 | 0.999 | 0.369 |
| Fear of earthquakes | 5.03 | 4.93 | 4.90 | 428 | 0.338 | 0.714 |
| Worries over buildings collapsing | 4.58 | 4.72 | 4.47 | 428 | 1.579 | 0.207 |
| Willingness to retrofit houses | 6.18 | 6.31 | 6.51 | 428 | 1.361 | 0.258 |
| Willingness to retrofit houses after assessment | 6.80 | 6.58 | 6.75 | 428 | 1.889 | 0.152 |
| Items | Postearthquake | | | | | |
| | Elementary/ Junior High | Senior High | University/ Graduate | DF | F | P-value |
| Probability of an earthquake disaster occurring within the next ten years | 4.25 | 4.67 | 4.83 | 469 | 7.468 | 0.001** |
| Fear of earthquakes | 5.12 | 5.28 | 5.26 | 469 | 0.402 | 0.669 |
| Worries over buildings collapsing | 4.64 | 5.07 | 4.69 | 469 | 3.100 | 0.046* |
| Willingness to retrofit houses | 4.12 | 4.19 | 4.22 | 469 | 0.154 | 0.857 |
| Willingness to retrofit houses after assessment | 5.27 | 5.45 | 5.69 | 469 | 5.342 | 0.005** |

* $p < 0.05$; ** $p < 0.01$; ***$p < 0.001$.

**Table 6** Post hoc results for education.

| Items | Education | Education | Mean Difference | Std. Error | Sig. | 95% Confidence Interval | |
|---|---|---|---|---|---|---|---|
| | | | | | | Lower Bound | Upper Bound |
| Probability of an earthquake disaster occurring within the next ten years (Postearthquake) Hochberg test | Elementary/Junior High | High School | -0.414 | 0.148 | 0.015* | -0.77 | -0.06 |
| | Elementary/Junior High | University/Graduate | -0.579 | 0.151 | 0.000*** | -0.94 | -0.22 |
| Retrofitting houses after professional assessment (Postearthquake) Hochberg test | Elementary/Junior High | University/Graduate | -0.420 | 0.133 | 0.005** | -0.74 | -0.10 |

$* p < 0.05$; $** p < 0.01$; $***p < 0.001$. $0.000*** \rightarrow 0.000142$

**Table 7** P-values and means for occupation.

| Items | Preearthquake | | | | | | |
|---|---|---|---|---|---|---|---|
| | Students | Home Makers | White-collar Workers | Blue-collar Workers | DF | F | P-value |
| Probability of an earthquake disaster occurring within the next ten years | 4.05 | 3.87 | 4.10 | 4.01 | 427 | 0.654 | 0.581 |
| Fear of earthquakes | 4.69 | 5.49 | 4.81 | 4.97 | 427 | 4.430 | 0.004** |
| Worries over buildings collapsing | 4.28 | 5.19 | 4.52 | 4.61 | 427 | 4.340 | 0.005** |
| Willingness to retrofit houses | 6.41 | 5.68 | 6.39 | 6.46 | 427 | 3.413 | 0.118 |
| Willingness to retrofit houses after assessment | 6.68 | 6.27 | 6.76 | 6.71 | 427 | 2.795 | 0.40 |

| Items | Postearthquake | | | | | | |
|---|---|---|---|---|---|---|---|
| | Students | Home Makers | White-collar Workers | Blue-collar Workers | DF | F | P-value |
| Probability of an earthquake disaster occurring within the next ten years | 4.35 | 4.82 | 4.80 | 4.47 | 469 | 3.475 | 0.016* |
| Fear of earthquakes | 4.47 | 6.04 | 5.16 | 5.06 | 469 | 12.266 | 0.000[1]*** |
| Worries over buildings collapsing | 4.26 | 5.25 | 4.88 | 4.72 | 469 | 3.392 | 0.018** |
| Willingness to retrofit houses | 4.32 | 4.34 | 4.41 | 3.91 | 469 | 3.995 | 0.008** |
| Willingness to retrofit houses after assessment | 5.38 | 5.57 | 5.67 | 5.35 | 469 | 2.873 | 0.036* |

[1]0.000000

* $p < 0.05$; ** $p < 0.01$; ***$p < 0.001$. 0.000***

**Table 8** Post hoc results for occupation

| Items | Education | Occupation | Mean Difference | Std. Error | Sig. | 95% Confidence Interval | |
|---|---|---|---|---|---|---|---|
| | | | | | | Lower Bound | Upper Bound |
| Fear of earthquakes (preearthquake) Hochberg test | Students | Home Makers | -0.797 | 0.263 | 0.015* | -1.49 | -0.10 |
| | Home Makers | White-collar Workers | 0.681 | 0.201 | 0.005** | 0.15 | 1.21 |
| Worries over buildings collapsing (preearthquake) Hochberg Test | Students | Home Makers | -0.909 | 0.277 | 0.007** | -1.64 | -0.18 |
| | Home Makers | White-collar Workers | 0.667 | 0.212 | 0.010* | 0.11 | 1.23 |
| | Home Makers | Blue-collar Workers | 0.586 | 0.209 | 0.032* | 0.03 | 1.14 |
| Fear of earthquakes (postearthquake) Games-Howell test | Students | Home Makers | -1.574 | 0.253 | 0.000***a | -2.24 | -0.90 |
| | Students | White-collar Workers | -0.693 | 0.254 | 0.041* | -1.37 | -0.02 |
| | Home Makers | White-collar Workers | 0.882 | 0.177 | 0.000***b | 0.42 | 1.34 |
| | Home Makers | Blue-collar Workers | 0.983 | 0.171 | 0.000***c | 0.54 | 1,43 |
| Worries over buildings collapsing (postearthquake) Games-Howell test | Students | Home Makers | -0.982 | 0.285 | 0.005** | -1.73 | -0.24 |
| Willingness to retrofit houses (postearthquake) Hochberg test | White-collar Workers | Blue-collar Workers | 0.499 | 0.156 | 0.009** | 0.09 | 0.91 |
| Willingness to retrofit houses after professional assessment (postearthquake) Hochberg test | White-collar Workers | Blue-collar Workers | 0.323 | 0.115 | 0.027* | 0.03 | 0.62 |

* $p < 0.05$; ** $p < 0.01$; ***$p < 0.001$. 0.000***a → 0.000000; 0.000***b → 0.00008; 0.000***c → 0.000000


**Table 9** P-values and means for house ownership.

| Items | Preearthquake | | | | | |
| --- | --- | --- | --- | --- | --- | --- |
| | Self-owned | Family-owned | Renting | DF | F | P-value |
| Probability of an earthquake disaster occurring within the next ten years | 4.02 | 4.09 | 3.98 | 427 | 0.317 | 0.728 |
| Fear of earthquakes | 5.05 | 4.89 | 4.74 | 427 | 2.087 | 0.125 |
| Worries over buildings collapsing | 4.64 | 4.65 | 4.46 | 427 | 0.642 | 0.527 |
| Willingness to retrofit houses | 6.25 | 6.43 | 6.44 | 427 | 0.806 | 0.447 |
| Willingness to retrofit houses after assessment | 6.75 | 6.57 | 6.66 | 427 | 1.248 | 0.288 |
| Items | Postearthquake | | | | | |
| | Self-owned | Family-owned | Renting | DF | F | P-value |
| Probability of an earthquake disaster occurring within the next ten years | 4.61 | 4.74 | 4.32 | 469 | 1.254 | 0.286 |
| Fear of earthquakes | 5.21 | 5.34 | 4.84 | 469 | 0.929 | 0.396 |
| Worries over buildings collapsing | 4.83 | 4.93 | 4.16 | 469 | 1.727 | 0.179 |
| Willingness to retrofit houses | 4.03 | 4.45 | 4.47 | 469 | 4.720 | 0.009** |
| Willingness to retrofit houses after assessment | 5.47 | 5.55 | 5.63 | 469 | 0.410 | 0.664 |

* $p < 0.05$; ** $p < 0.01$; ***$p < 0.001$.

**Table 10** Post hoc results for house ownership.

| Items | Education | House Ownership | Mean Difference | Std. Error | Sig. | 95% Confidence Interval | |
| --- | --- | --- | --- | --- | --- | --- | --- |
| | | | | | | Lower Bound | Upper Bound |
| Willingness to retrofit houses (postearthquake) Hochberg test | Self-owned | Family-owned | -0.424 | 0.144 | 0.014* | -0.78 | -0.07 |
| | Family-owned | Self-owned | 0.424 | 0.144 | 0.014* | 0.07 | 0.78 |

* $p < 0.05$; ** $p < 0.01$; ***$p < 0.001$.