# Peer review of "Exploring the Changes in Risk Perceptions and Adaptation Behaviors Based on Various Socioeconomic Characteristics Before and After Earthquake Disasters – A Case Study in Taiwan"

_Natural Hazards and Earth System Sciences, 2019_

## Referee Comment (RC1) · Anonymous Referee #1 · 23 Mar 2020

Summary: As the title indicates, the objective of this manuscript is to explain changes in seismic risk perception and adaptation behavior after an earthquake among different demographic groups. The literature review references a number of relevant citations but also cites tangentially related and outdated citations and overlooks two important reviews and some very relevant recent citations (see the list below). The Introduction fails to state specific research questions or research hypotheses. The data set appears to be excellent but the procedures for sampling cases and measuring items are inadequately described. The Results section is subdivided by the major demographic

variables, but those headings don't accurately describe the presentation of results— some of which are about pretest-posttest differences that appear to be unrelated to the demographic variables. Moreover, the results are presented in a series of unconventional figures that fail to provide the reader with adequate information about the effects sizes for the impact of the demographic variables on the dependent variables or correlations among dependent variables. The Discussion and Conclusions focus on the effects of the demographic variables on risk perception and adaptation behavior but ignore the pretest-posttest differences. This is a significant limitation because these sections fail to address a major part of the study's stated objective. In addition, systematic reviews of the disaster research literature indicate that demographic variables have small and inconsistent effects on adaptation behavior, so the authors are probably focusing on the least important part of their study's results. Finally, as a general comment, I know from personal experience how difficult it is to submit papers that is not written in my native language. Accordingly, I seek the assistance of a professional editor before submitting papers in other languages. The authors of this manuscript should have done this already and should definitely do so before resubmission.

Line Comment 44 The section on risk perception cites literature that is either overly general (Eagly and Chaiken, 1993, is about attitudes rather than risk perception) or outdated (Sjöberg, 2000; Sjöberg, 1996). Moreover, although risk perception might be influenced by internal and external factors, it does not "sum up" those factors.

64 The title makes it reasonably clear what are the study's research objectives, but there is no clear statement of research questions or research hypotheses at the conclusion of the Introduction. This might be why the Results and Discussion sections fail to adequately describe the changes in risk perception and adaptation behavior.

78 Figure 1a is sufficient for a research article. Figure 1b, 1c, and 1d are only of interest to local authorities.

89 It is unclear what it meant by "simple random sampling". Is this simple random sampling from a sample frame (i.e., a list of telephone numbers) or random digit dialing?

99 The section describing the measures should not be referring to the research literature. Those references should have already been cited in the Introduction's literature review. Instead, this section should specifically describe each item in the questionnaire and how it was measured. Thus, the description of the items "probability of an earthquake disaster occurring within ten years", "fear of earthquake", and "worry of building collapse" should list the exact English translation of those items and list the rating scale anchors that were used (e.g., "Not at all = 1 to Almost a certainty = 7" for the earthquake probability rating). The items measuring "the impacts they expected from the disaster" should be replaced by a statement of the specific impacts that were listed.

114 Most of the first paragraph in this section is, or should be, common knowledge among survey researchers. Consequently, all but the last sentence should be deleted—as should Figure 2.

142 Table 1 should also contain data for the distributions of gender, age, education, occupation, and homeownership for the study area so readers can assess the extent of sample bias.

144 Section 3.1 is labeled sex but presents a number of results that appear to be unrelated to sex differences. Specifically, "the earthquake probability (the P value of 0.049), the fear =of earthquake (the P value of 0.000), and the willingness on house retrofit (the P value of 0.002) are statistical significance indicating a serious earthquake indeed increase awareness of disaster" seems to be a pretest-posttest comparison that is unrelated to sex differences. This problem continues throughout the rest of the Results section.

148 Figure 3 presents the results in a format that is rather inventive, but extremely confusing and relatively uninformative, compared to the conventional method of presenting a matrix containing the variables' means in the first column, the standard deviations in the second column, and the intercorrelations in the remaining columns. In addition

providing effect sizes for to the impact of the independent variables on the dependent variables, a correlation matrix allows the reader to see the correlations among the dependent variables (see Lindell & Hwang, 2008, for an example). Providing this correlation matrix will eliminate the need for Figures 4-7, as well.

191 The Discussion section only addresses the effects of the demographic variables, ignoring the effects of changes in risk perception and their possible effects on risk reduction actions.

200 Figures 8-10 might be useful for guiding local officials' hazard awareness programs, but they do not contribute to general scientific knowledge. Thus, they should be deleted.

References J.S. Becker, D. Paton, D.M. Johnston, K.R. Ronan, A model of household preparedness for earthquakes: how individuals make meaning of earthquake information and how this influences preparedness, Nat. Hazards. 64 (2012) 107–137. J.S. Becker, D. Paton, D.M. Johnston, K.R. Ronan, J. McClure, The role of prior experience in informing and motivating earthquake preparedness, Int. J. Disaster Risk Reduct. 22 (2017) 179–193. J.S. Becker, S.H. Potter, S.K. McBride, A. Wein, E.E.H. Doyle, D. Paton, When the earth doesn't stop shaking: How experiences over time influenced information needs, communication, and interpretation of aftershock information during the Canterbury Earthquake Sequence, New Zealand, Int. J. Disaster Risk Reduct. 34 (2019) 397–411. E.E.H. Doyle, J. McClure, S.H. Potter, J.S. Becker, D.M. Johnston, M.K. Lindell, S. Johal, S.A. Fraser, M.A. Coomer, Motivations to prepare after the 2013 Cook Strait Earthquake, N.Z., Int. J. Disaster Risk Reduct. 31 (2018) 637–649. M.K. Lindell. North American cities at risk: Household responses to environmental hazards. In H. Joffe, T. Rossetto & J. Adams (Eds.). Cities at Risk: Living with Perils in the 21st Century. Springer, Dordrecht, 2013:109-130. M.K. Lindell, S.N. Hwang, Households' perceived personal risk and responses in a multihazard environment. Risk Anal. 28 (2008) 539-556. C. Solberg, T. Rossetto, H. Joffe. The social psychology of seismic hazard adjustment: Re-evaluating the international literature. Nat. Haz. Earth Sys.

Sci. 10 (2010) 1663-1677.

---

## Referee Comment (RC2) · Anonymous Referee #2 · 17 Apr 2020

The piece of the change of risk perception and adaptation behavior between pre- and post-earthquake disaster proposes an interesting comparative discussion. The manuscript has a clear scope but some sections could be improved. In addition, there are some other literature exploring similar topics (listed below) and should be included in the discussion. Indeed, risk perception and adaptive actions might be varied according to different social characters. The presentation of result is radical different from previous studies in ANOVA. Traditional table could reveal various value and significance. Authors should provide more information of such different expression to let

reader catch such outcome. As a whole, the dataset is interesting and meaningful for most studies indeed could only examine pre- or post- earthquake only. In the following I would like to separate my comments into general and specific.

(1) General comments A.Although risk perception and adaptation behavior are the key issue, it seems that disaster experience is the key factor authors discussed in this article. The overall logic in introduction is blurred right now, and such vague might further the results interpretation. How to reconnect the research question and the findings might be important for this study.

B.The expression for the results need more information. It is easy for readers to catch the results from table such as the value of the variable and the p-value. Although in Figure 3 to 7 there is a red line for p-value of 0.05, the figures are still blurred. What does the arrow mean? In order to increase readability, certain information might be necessary to provide.

(2) Specific comments A.Line 35. Current reference applied to risk perception and adaptation behavior is rather too old. In fact, there are more recent literature exploring similar issues or topics. Although some of the literature are important such as Lindell, Becker, Sjöberg and so on, it is important to update such discussion.

Motivations to prepare after the 2013 Cook Strait Earthquake, N.Z Perceptions and reactions to tornado warning polygons: Would a gradient polygon be useful? Assessment of households' responses to the tsunami threat: A comparative study of Japan and New Zealand Perceptions, behavioral expectations, and implementation timing for response actions in a hurricane emergency Port stakeholder perceptions of Sandy impacts: a case study of Red Hook, New York Conflicts in adaptation: case studies from Nepal and the Maldives The role of prior experience in informing and motivating earthquake preparedness

B.Line 51. The research question might need more specific and elaborated in the last paragraph of Introduction section. Although the title is rather clear, there is no

statement regarding the research question. Therefore, this part could be improved.

C.Line 85. In the article, the survey data is the main dataset. "All survey sampling methods relied on simple random sampling." How can you tell the representative of the sampling data? What is the ratio between sampling amount and the study area?

D.Figure 2 is important for this study. However, it is unclear which result is applied post hoc or not. This should be discussed systematically either in the research design or in the results.

E.Line 135. The separation of the result is based upon social character. Again, due to there is no specific research question, it is hard for readers to understand why separate in current sub-categories. In addition, I think pre- and post- is the main concern, and this should be clarified.

F.Line 191. Figure 8 to 10 are not providing enough information for the readers, so I suggest these figure could be deleted.

---

## Referee Comment (RC3) · Anonymous Referee #3 · 21 Apr 2020

This is an interesting study investigating risk perception and preparedness actions through surveys, pre and post an earthquake event. The findings are important and contribute to the growing body of research in this space. However, currently this manuscript requires significant revisions for those findings to be recognised clearly. Primarily, areas of research are missing in the introduction and discussion, that both can provide more context and help the authors interpret some of their findings. Second, there is a lack of clarity in a number of places, including the presentation of methodology, results, and figures. Thirdly, while I appreciate the challenge of writing in a

second language (and acknowledge the privilege of being able to write in my first), the manuscript is currently very difficult to read and understand in places – which sadly detracts from the data within. I would recommend the editors consult a professional editing service, seek assistance from the journal if it offers that option, or seek an additional author to assist with the writing. My detailed recommendations follow below,

Some substantial areas of research are missing, including key preparedness and response literature. For example, discussion of the Protective Action Decision Model (Lindell & Perry, 2012; Lindell & Hwang 2008; Lindell & Prater 2002) and other hazard preparedness models (e.g., Paton et al., 2015), societal influences on household preparedness (Becker et al 2014), recent research on preparedness motivations (Becker et al 2017; Doyle et al 2018) which consider multiple-event risk perceptions (McClure et al 2016, Doyle et al 2018). Thus, the introduction and contextual discussion of the findings should explore some of the elements raised in this pre-existing literature, and how the results relate to those, including self- and collective-efficacy, outcome expectancy, responsibility, etc. Notably, the discussion omits a number of key texts investigating the relationship between gender and preparedness, and reasons for that difference (e.g., familial responsibilities), e.g. Dooley et al; Bateman and Edwards; Lindell & Prater; Olofsson & Rashid; Palmer; as well as texts exploring the barriers to preparedness (e.g., Blake et al 2018; Senkbeil et al 2014)

The Literature on attribution theory, and on trust in communications is also lacking.

The authors need to set the scene more in section 2.1 (Study Area) – what resilience building activities have been conducted in these regions, if at all? Some more information about the community, and previous events or resilience activities is needed.

Section 2.2 – 'simple random sampling' – of what? The phone records?

Section 2.2 – what do you mean by 'some notifications'?

Section 2.2 – for the survey questions, why did you choose these particular factors

(e.g., trust in government and responsibility attribution?). These factors need more detail explanation in the introduction, referencing the relevant literature (e.g., on trust and attribution theory for risk communication), such that in section 2.2 there is more rationale and explanation for their choice (and prioritisation over other potential factors).

I do not think the authors need to include the full description of the ANOVA in section 2.4, given it is a well know statistical test. Would recommend trimming, or if needed including in an appendix. Much of section 2.4 can be summarised much more briefly, as these are standard approaches.

The tale end of section 2.4, outlining the Likert scale of 1 to 7, should be moved to section 2.3 where the measures are discussed.

Lines 145 to 150, seems to contradict oneself on first reading – first you say that there is willingness to house retrofit, and then that it decreased. I think you mean pre/post earthquake, but this needs clarification.

In general the ANOVA results need clearer reporting, to standard (brief style) formatting including more clearly the F statistic and degrees of freedom, rather than just the P value. To that end, the P value isn't 0.000, but should be reported as $p < 0.0005$.

Line 140, explain why high school/blue collar might have less capability to adjust in the introduction, to set the context here.

Section 3.2 (Age) is hard to follow, and needs rewording completely, as currently it reads contradictory.

Line 190 – this should be explored in the discussion, in the context of how both resources (\$) and / or care responsibilities could be a possible interpretation of this finding.

Section 3.3 – does the higher education group correlate with income? Equally, could this group have less worry of buildings collapsing, because they could afford to have better buildings to start with (or were able to retrofit)? These nuances need further

discussion.

Line 193 – This discussion of gender needs more explanation in context of the references listed previously.

Throughout the figures need more context and linking to the text (e.g., Figure 8 on line 200 – it's hard to link the text to the figure. The figures need more explanatory captions to guide the reader, and the text needs more explanatory linking to the figures.

Other issues (such as fatalism, or anxiety) need to be raised in the discussion in more detail – see e.g., McClure et al 2001 and Paton 2005, Wei & Lindell 2017

Line 227 – I'm not sure where the concept of bounded rationality came from in this paragraph. Needs better linking and explanation.

Line 229 – what do you mean by internal control?

Line 228-231 repeats exactly some sentences in the introduction – reword appropriately

The discussion needs some more explanation of the limitations – a limitation and future research section would be ideal. They are touched on in the conclusion (e.g., time limitation), but lack enough detail for the reader to evaluate and interpret.

General comments

The authors introduce the term 'subjective resilience' but need to define and explain this further. How does it relate to the various measures discussed?

The authors refer to 'sex' (see section 3.1 in particular), when I believe they need to be referring to 'gender' as the issues here relate to familial responsibilities and social roles relating to someone's gender – not their biological sex. See Rushton et al (2019) for more.

Figures need improving for clarity. Figure 1a-d need to be larger as the keys are hard

to read, Figure 2 would be better in an appendix. All figures would benefit from more explanatory extensive captions that enable them to be read and interpreted more easily. Figures 3, 4, 5, 6 & 7 are not in a format I am familiar with. It took a while to interpret them, I feel they need some much clearer captions and further explanation to facilitate interpretation.

Some of the tables are a bit unclear, e.g., Table 2: it is hard to follow which of the rows in column 1 apply to which rows in the other columns. Can they be reformatted to aid comprehension?

References ========

Rushton A, Gray L, Canty J, & Blanchard K (2019) Beyond binary: (re)defining "gender" for 21st century disaster risk reduction research, policy, and practice Int J Environ Res Public Health, https://doi.org/10.3390/ijerph16203984.

McClure J, Allen MW, & Walkey F (2001) Countering fatalism: Causal information in news reports affects judgments about earthquake damage Basic Appl Soc Psych, https://doi.org/10.1207/S15324834BASP2302_3.

Paton D, Smith L, & Johnston D (2005) When good intentions turn bad : promoting natural hazard preparedness Aust J Emerg Manag 20(1) 25–30.

Wei H-L, & Lindell MK (2017) Washington households' expected responses to lahar threat from Mt. Rainier Int J Disaster Risk Reduct 22 77–94, https://doi.org/10.1016/J.IJDRR.2016.10.014.

Senkbeil JC, Scott DA, Guinazu-Walker P, & Rockman MS (2014) Ethnic and Racial Differences in Tornado Hazard Perception, Preparedness, and Shelter Lead Time in Tuscaloosa Prof Geogr 66(4) 610–620, https://doi.org/10.1080/00330124.2013.826562.

Blake D, Marlowe J, & Johnston D (2017) Get prepared: Discourse for the privileged? Int J Disaster Risk Reduct 25 283–288, https://doi.org/10.1016/J.IJDRR.2017.09.012.

Olofsson A, & Rashid S (2011) The White (Male) Effect and Risk Perception: Can Equality Make a Difference? Risk Anal 31(6) 1016–1032, https://doi.org/10.1111/j.1539-6924.2010.01566.x.

Lindell MK, & Prater C (2002) Risk area resident' perceptions and adoption of seismic hazard adjustments J Appl Soc Psychol 32(11) 2377–2392.

Palmer CGS (2003) Risk perception: Another look at the "white male" effect Heal Risk Soc 5(1) 71–83, https://doi.org/10.1080/1369857031000066014.

Dooley D, Catalano R, Mishra S, & Serxner S (1992) Earthquake Preparedness: Predictors in a Community Survey J Appl Soc Psychol 22(6) 451–470, https://doi.org/10.1111/j.1559-1816.1992.tb00984.x.

Bateman JM, & Edwards B (2002) Gender and Evacuation: A Closer Look at Why Women Are More Likely to Evacuate for Hurricanes Nat Hazards Rev 3(3) 107–117, https://doi.org/10.1061/(ASCE)1527-6988(2002)3:3(107).

Lindell MK, & Perry RW (2012) The Protective Action Decision Model: Theoretical Modifications and Additional Evidence Risk Anal 32(4) 616–632, https://doi.org/10.1111/j.1539-6924.2011.01647.x.

Becker JS, Paton D, & Johnston DM (2014) Societal Influences on Earthquake Information Meaning-Making and Household Preparedness Int J Mass Emerg Disasters 32(2) 317–352.

Lindell MK, & Hwang SN (2008) Households ' Perceived Personal Risk and Responses in a Multihazard Environment , https://doi.org/10.1111/j.1539-6924.2008.01032.x.

Lindell MK, & Prater C (2002) Risk area resident' perceptions and adoption of seismic hazard adjustments J Appl Soc Psychol 32(11) 2377–2392.

Becker JS, Paton D, Johnston DM, Ronan KR, & McClure J (2017) The role of prior experience in informing and motivating earthquake preparedness Int J Disaster Risk

Reduct 22 179–193, https://doi.org/10.1016/J.IJDRR.2017.03.006.

McClure J, Henrich L, Johnston DM, & Doyle EEH (2016) Are two earthquakes better than one? How earthquakes in two different regions affect risk judgments and preparation in three locations Int J Disaster Risk Reduct 16 192–199, https://doi.org/10.1016/j.ijdrr.2016.03.003.

Paton D, Anderson E, Becker J, & Petersen J (2015) Developing a comprehensive model of hazard preparedness: Lessons from the Christchurch earthquake Int J Disaster Risk Reduct 14 37–45, https://doi.org/10.1016/j.ijdrr.2014.11.011.

Doyle EEH, McClure J, Potter SH, Becker JS, Johnston DM, Lindell MK, Johal S, Fraser SA, & Coomer MA (2018) Motivations to prepare after the 2013 Cook Strait Earthquake, N.Z. Int J Disaster Risk Reduct 31 637–649, https://doi.org/10.1016/j.ijdrr.2018.07.008.
* * *

---

## Author Comment (AC1) · 18 May 2020

As the title indicates, the objective of this manuscript is to explain changes in seismic risk perception and adaptation behavior after an earthquake among different demographic groups. The literature review references a number of relevant citations but also cites tangentially related and outdated citations and overlooks two important reviews and some very relevant recent citations (see the list below). The Introduction fails to state specific research questions or research hypotheses. The data set appears to be excellent but the procedures for sampling cases and measuring items are inadequately

described. The Results section is subdivided by the major demographic variables, but those headings don't accurately describe the presentation of results some of which are about pretest-posttest differences that appear to be unrelated to the demographic variables. Moreover, the results are presented in a series of unconventional figures that fail to provide the reader with adequate information about the effects sizes for the impact of the demographic variables on the dependent variables or correlations among dependent variables. The Discussion and Conclusions focus on the effects of the demographic variables on risk perception and adaptation behavior but ignore the pretest-posttest differences. This is a significant limitation because these sections fail to address a major part of the study's stated objective. In addition, systematic reviews of the disaster research literature indicate that demographic variables have small and inconsistent effects on adaptation behavior, so the authors are probably focusing on the least important part of their study's results. Finally, as a general comment, I know from personal experience how difficult it is to submit papers that is not written in my native language. Accordingly, I seek the assistance of a professional editor before sub-mitting papers in other languages. The authors of this manuscript should have done this already and should definitely do so before resubmission.

Ans: Thanks for your general and specific comments, and they have great help on im-proving the research. Indeed English is not our native language, and many thanks for your recommendation. In fact, this paper has been submitted for English proofreading before submitting Natural Hazards and Earth System Sciences. We have transferred your valuable comments to American Journal Expert and the resubmission will be reed-ited again by native English speakers.

The followings are the point-by-point responses.

1.Line Comment 44 The section on risk perception cites literature that is either overly general (Eagly and Chaiken, 1993, is about attitudes rather than risk perception) or outdated (Sjöberg, 2000; Sjöberg, 1996). Moreover, although risk perception might be influenced by internal and external factors, it does not "sum up" those factors.

[Figure]

Ans: Thanks for the comment. The purpose of this article is to explore the change of risk perception and adaptation behavior between the pre- and the post-earthquake. In order to identify main research topic, the revised version will improve both risk perception and the potential influence of disaster experience according to the comments.

2.64 The title makes it reasonably clear what are the study's research objectives, but there is no clear statement of research questions or research hypotheses at the conclusion of the Introduction. This might be why the Results and Discussion sections fail to adequately describe the changes in risk perception and adaptation behavior.

Ans: Thanks for the insightful comment. As a whole, this study contributes to the exploration of how earthquake disasters influence the risk perception and adaptation behavior of residents in Taiwan and further categorizes according to the social characters. Based upon past studies, the interactions of social characters could collectively affect responses to disasters. This study will then discuss the response from various social characters respectively to explore how social characters affect the pre- and the post- risk perception and adaptation behavior. The revised version will then improve the statement of research questions in the Introduction and to further improve the consistency between the title and the article.

3.78 Figure 1a is sufficient for a research article. Figure 1b, 1c, and 1d are only of interest to local authorities.

Ans: Thanks for the comment. In order to leave accurate information, the revised version will delete the rest figures in Figure 1 according to the comments.

4.89 It is unclear what it meant by "simple random sampling". Is this simple random sampling from a sample frame (i.e., a list of telephone numbers) or random digit dialing?

Ans: This study adopted voluntary response sampling within the study area. In order to examine the variation of risk perception an adaptation behavior, the paper conducted

street survey before the earthquake and the telephone survey in the after. In order to clarify the sample collecting process, the revised version will improve the section of "data collection".

5.99 The section describing the measures should not be referring to the research literature. Those references should have already been cited in the Introduction's literature review. Instead, this section should specifically describe each item in the questionnaire and how it was measured. Thus, the description of the items "probability of an earthquake disaster occurring within ten years", "fear of earthquake", and "worry of building collapse" should list the exact English translation of those items and list the rating scale anchors that were used (e.g., "Not at all = 1 to Almost a certainty = 7" for the earthquake probability rating). The items measuring "the impacts they expected from the disaster" should be replaced by a statement of the specific impacts that were listed.

Ans: Thanks for the comment. The purpose of section 2.3 is to illustrate the survey questions in the study. In order to separate the data and literature review, the updated version has revised this section and focus on explaining the variables used. In addition, the revised version will add up new Table 1 to explain the measurement of questionnaires.

6.114 Most of the first paragraph in this section is, or should be, common knowledge among survey researchers. Consequently, all but the last sentence should be deleted as should Figure 2.

Ans: Thanks for the comment. The first paragraph in section 2.4 is to give a general concept of ANOVA to the readers. However, it is indeed a common knowledge among survey researchers. Therefore, the revised version has deleted the first sentence for it is too general but keep the second sentence regarding one-way analysis of variance. In addition, Figure 2 is deleted in the revised version as well.

7.142 Table 1 should also contain data for the distributions of gender, age, education, occupation, and homeownership for the study area so readers can assess the extent

of sample bias.

Ans: Thanks for the comment. The table could be further improved to present the distributions of both sample and the study area. Therefore, the revised version will collect relevant data for readers to assess the extent of sample biass.

8.144 Section 3.1 is labeled sex but presents a number of results that appear to be unrelated to sex differences. Specifically, "the earthquake probability (the P value of 0.049), the fear =of earthquake (the P value of 0.000), and the willingness on house retrofit (the P value of 0.002) are statistical significance indicating a serious earthquake indeed increase awareness of disaster" seems to be a pretest-posttest comparison that is unrelated to sex differences. This problem continues throughout the rest of the Results section.

Ans: Thanks for the valuable comment. The purpose of this article is to explore the change of risk perception and adaptation behavior among varied social character between the pre- and the post- earthquake. Therefore, the revised version has emphasized such discussion in the result section.

9.148 Figure 3 presents the results in a format that is rather inventive, but extremely confusing and relatively uninformative, compared to the conventional method of presenting a matrix containing the variables' means in the first column, the standard deviations in the second column, and the intercorrelations in the remaining columns. In addition providing effect sizes for to the impact of the independent variables on the dependent variables, a correlation matrix allows the reader to see the correlations among the dependent variables (see Lindell & Hwang, 2008, for an example). Providing this correlation matrix will eliminate the need for Figures 4-7, as well.

Ans: Thanks for the comment. Due to the purpose is to compare the change between various social character and the time period, the arrows and the lines are used to express such outcome. However, like reviewer mentioned, the figure might not a perfect way to present the results and make it more confusing. Therefore, the conventional

tables will be applied to show the overall results among social characters.

10.191 The Discussion section only addresses the effects of the demographic variables, ignoring the effects of changes in risk perception and their possible effects on risk reduction actions.

Ans: Thanks for the comment. Based upon past studies, the interactions of social characters could collectively affect responses to disasters. Therefore, the purpose of this article is to discuss the response from various social characters respectively to explore how social characters affect the pre- and the post- risk perception and adaptation behavior. The revised version will improve the discussion section of the potential impacts on the change of disaster perception and adaptive behavior from the interactions of social characters.

---

## Author Comment (AC2) · 18 May 2020

The piece of the change of risk perception and adaptation behavior between pre and post-earthquake disaster proposes an interesting comparative discussion. The manuscript has a clear scope but some sections could be improved. In addition, there are some other literature exploring similar topics (listed below) and should be included in the discussion. Indeed, risk perception and adaptive actions might be varied according to different social characters. The presentation of result is radical different from previous studies in ANOVA. Traditional table could reveal various value and sig-

nificance. Authors should provide more information of such different expression to let reader catch such outcome. As a whole, the dataset is interesting and meaningful for most studies indeed could only examine pre- or post- earthquake only.

Ans: Thanks for your general and specific comments, and they have great help on improving the research. First of all, thanks for providing related references for this article, and the revised version will then include certain works. It seems that the current presentation of the results might confuse readers, and the revised version will take the comments into account to improve such confusion.

In the following I would like to separate my comments into general and specific.

1.Although risk perception and adaptation behavior are the key issue, it seems that disaster experience is the key factor authors discussed in this article. The overall logic in introduction is blurred right now, and such vague might further the results interpretation. How to reconnect the research question and the findings might be important for this study.

Ans: Thanks for the comment. The study attempts to discuss the change of risk perception and adaptation behavior among varied social character among pre- and post-earthquake disaster. The research question is not clear enough in the current version, and the revised version will improve such statement in both "Introduction" and "Conclusion." The clear research question might help to reconnect the motivation and the findings.

2.The expression for the results need more information. It is easy for readers to catch the results from table such as the value of the variable and the p-value. Although in Figure 3 to 7 there is a red line for p-value of 0.05, the figures are still blurred. What does the arrow mean? In order to increase readability, certain information might be necessary to provide.

Ans: Thanks for the comment. The arrow in Figure 3 to 7 indicates the change of the

disaster perception and adaptive behaviors. The current presentation is confusing, and the revised version will present the findings based upon tradition ANOVA to clarify the results.

3.Line 35. Current reference applied to risk perception and adaptation behavior is rather too old. In fact, there are more recent literature exploring similar issues or topics. Although some of the literature are important such as Lindell, Becker, Sjöberg and so on, it is important to update such discussion. Motivations to prepare after the 2013 Cook Strait Earthquake, N.Z Perceptions and reactions to tornado warning polygons: Would a gradient polygon be useful? Assessment of households' responses to the tsunami threat: A comparative study of Japan and New Zealand Perceptions, behavioral expectations, and implementation timing for response actions in a hurricane emergency Port stakeholder perceptions of Sandy impacts: a case study of Red Hook, New York Conflicts in adaptation: case studies from Nepal and the Maldives The role of prior experience in informing and motivating earthquake preparedness

Ans: Thanks for the comment. The revised version will take the suggested references into consideration and improve the statement.

4.Line 51. The research question might need more specific and elaborated in the last paragraph of Introduction section. Although the title is rather clear, there is no statement regarding the research question. Therefore, this part could be improved.

Ans: Thanks for the comment. The revised version will add up the research questions in both introduction and conclusion to improve the overall logic in the study.

5.Line 85. In the article, the survey data is the main dataset. "All survey sampling methods relied on simple random sampling." How can you tell the representative of the sampling data? What is the ratio between sampling amount and the study area?

Ans: Thanks for the comment. Regarding the representative of the sampling data, Table 1 will contain data for the distributions of gender, age, education, occupation,

and homeownership for both the study area and the sampling so readers can assess the extent of sample bias.

6.Figure 2 is important for this study. However, it is unclear which result is applied post hoc or not. This should be discussed systematically either in the research design or in the results.

Ans: Thanks for the comment. The revised version will improve Figure 2 and the application of post hoc in the results.

7.Line 135. The separation of the result is based upon social character. Again, due to there is no specific research question, it is hard for readers to understand why separate in current sub-categories. In addition, I think pre- and post- is the main concern, and this should be clarified.

Ans: Thanks for the comment. Indeed, the main concerns are social character and pre- and post-. Therefore, the revised version will state clearly regarding the research question and to make it become more consistency through the whole study.

8.Line 191. Figure 8 to 10 are not providing enough information for the readers, so I suggest these figure could be deleted.

Ans: Thanks for the comment. Originally, the purpose of Figure 8 to 10 is for future disaster management by taking into social character into account. However, the Figures indeed might not provide enough information for now, and the revised version will then delete them.

---

## Author Comment (AC3) · 18 May 2020

This is an interesting study investigating risk perception and preparedness actions through surveys, pre and post an earthquake event. The findings are important and contribute to the growing body of research in this space. However, currently this manuscript requires significant revisions for those findings to be recognised clearly. Primarily, areas of research are missing in the introduction and discussion, that both can provide more context and help the authors interpret some of their findings. Second, there is a lack of clarity in a number of places, including the presentation of methodology, results, and figures. Thirdly, while I appreciate the challenge of writing in a second language (and acknowledge the privilege of being able to write in my first), the manuscript is currently very difficult to read and understand in places – which sadly detracts from the data within. I would recommend the editors consult a professional editing service, seek assistance from the journal if it offers that option, or seek an additional author to assist with the writing.

Ans: Thanks for the valuable and insightful comments. First of all, thanks for pointed out that there are areas of research missing in introduction and discussion, and the revised version will focus on these two section to improve both statements and interpretation. Secondly, the submitted paper has been through English proofreading in American Journal Experts, and the comments have been transferred to AJE and hopefully the revised version could solve the issue.

My detailed recommendations follow below,

1.Some substantial areas of research are missing, including key preparedness and response literature. For example, discussion of the Protective Action Decision Model (Lindell & Perry, 2012; Lindell & Hwang 2008; Lindell & Prater 2002) and other hazard preparedness models (e.g., Paton et al., 2015), societal influences on household preparedness (Becker et al 2014), recent research on preparedness motivations (Becker et al 2017; Doyle et al 2018) which consider multiple-event risk perceptions (McClure et al 2016, Doyle et al 2018). Thus, the introduction and contextual discussion of the findings should explore some of the elements raised in this pre-existing literature, and how the results relate to those, including self- and collective-efficacy, outcome expectancy, responsibility, etc. Notably, the discussion omits a number of key texts investigating the relationship between gender and preparedness, and reasons for that difference (e.g., familial responsibilities), e.g. Dooley et al; Bateman and Edwards; Lindell & Prater; Olofsson & Rashid; Palmer; as well as texts exploring the barriers to preparedness (e.g., Blake et al 2018; Senkbeil et al 2014). The Literature on attribution theory, and on trust in communications is also lacking.

Ans: Thanks for the valuable comments and providing abundant references for us. Based upon past studies, the interactions of social characters could collectively affect responses to disasters. This study discusses the response from various social characters respectively to explore how social characters affect the pre- and the post- risk perception and adaptation behavior. The revised version will take all the suggested areas of research into consideration and improve the statements in both introduction and discussion section.

2.The authors need to set the scene more in section 2.1 (Study Area) – what resilience building activities have been conducted in these regions, if at all? Some more information about the community, and previous events or resilience activities is needed.

Ans: Thanks for the comment. The revised version will include more information related to resilience achievements regarding the study area.

3.Section 2.2 – 'simple random sampling' – of what? The phone records?

Ans: Thanks for the comment. All survey sampling methods relied on voluntary response sampling. The former is streets survey, and the latter is telephone survey based on phone number databases within study area by the survey research center of the domestic academic institution. The revised version will improve the illustration in data collection.

4.Section 2.2 – what do you mean by 'some notifications'?

Ans: Here "some notification" indicates to give some information for the respondents such as when was last time of earthquake magnitude was over 6.0, where the fault line is, what is the frequency of earthquake disaster in the study area and so on. Current version has not stated clearly and the revised version will improve the statement in data collection.

5.Section 2.2 – for the survey questions, why did you choose these particular factors (e.g., trust in government and responsibility attribution?). These factors need more

detail explanation in the introduction, referencing the relevant literature (e.g., on trust and attribution theory for risk communication), such that in section 2.2 there is more rationale and explanation for their choice (and prioritisation over other potential factors).

Ans: Thanks for the comment. This study has reviewed the past research on the impact of social character and pre- and post- disaster. There are purposes of applying such questions to detect individuals' disaster perception and adaptive behaviors. The revised version will clarify the reasons and principles of the factors applied in this article.

6.I do not think the authors need to include the full description of the ANOVA in section 2.4, given it is a well know statistical test. Would recommend trimming, or if needed including in an appendix. Much of section 2.4 can be summarised much more briefly, as these are standard approaches.

Ans: Thanks for the comment. The revised version will give brief intro of ANOVA.

7.The tale end of section 2.4, outlining the Likert scale of 1 to 7, should be moved to section 2.3 where the measures are discussed.

Ans: Thanks for the comment. The revised version will move the Likert scale of 1 to 7 to section 2.3.

8.Lines 145 to 150, seems to contradict oneself on first reading – first you say that there is willingness to house retrofit, and then that it decreased. I think you mean pre/post earthquake, but this needs clarification.

Ans: Thanks for the comment. Due to the research question is not that clear in the introduction and further results in the confusing statement in the findings. The revised version will make it consistency to state both social character and pre- and post- disaster change.

9.In general the ANOVA results need clearer reporting, to standard (brief style) formatting including more clearly the F statistic and degrees of freedom, rather than just the

P value. To that end, the P value isn't 0.000, but should be reported as p<0.0005.

Ans: Thanks for the comment. The revised version will provide the traditional ANOVA to give a clear reports of the results.

10.Line 140, explain why high school/blue collar might have less capability to adjust in the introduction, to set the context here.

Ans: Thanks for the comment. Due to the different educational base, the capability might be varied. The revised version will bring more discussions first in the introduction section to explain such issue.

11.Section 3.2 (Age) is hard to follow, and needs rewording completely, as currently it reads contradictory.

Ans: Thanks for the comment, and the revised version will rewrite section 3 and 4.

12.Line 190 – this should be explored in the discussion, in the context of how both resources ($) and / or care responsibilities could be a possible interpretation of this finding.

Ans: Thanks for the comment. As we know, households with children could have more willingness to pay more for house retrofit, and self-owned household could believe original house safety or relied on their life experience. However, it is suitable for complete domestic research because of cultural difference and detailed socioeconomic information. That is the reason why we decide to have a comprehensive comparison instead of specific discussion. We believe it is a good issue for future work.

13.Section 3.3 – does the higher education group correlate with income? Equally, could this group have less worry of buildings collapsing, because they could afford to have better buildings to start with (or were able to retrofit)? These nuances need further discussion.

Ans: Thanks for the comment. Here, in the study area, there is a trend that higher

education might have relative higher income. The revised version will add up more discussions in the introduction section to lead more rational discussion in the discussion section.

14.Line 193 – This discussion of gender needs more explanation in context of the references listed previously.

Ans: Thanks for the comment and we will improve the whole discussion part in the revised version accordingly.

15.Throughout the figures need more context and linking to the text (e.g., Figure 8 on line 200 – it's hard to link the text to the figure. The figures need more explanatory captions to guide the reader, and the text needs more explanatory linking to the figures.

Ans: Thanks for the comment. Figure 8 to 10 could only provide limited information and therefore we will delete the figures in the revised version. In addition, the whole section of discussion will be rewritten to improve the overall research purpose.

16.Other issues (such as fatalism, or anxiety) need to be raised in the discussion in more detail – see e.g., McClure et al 2001 and Paton 2005, Wei & Lindell 2017

Ans: Thanks for the comment, and the revised version will focus on discussing the potential impacts of social character and pre- and post- disaster. Therefore, the research question will be stated in the introduction and other issues will be taken away.

17.Line 227 – I'm not sure where the concept of bounded rationality came from in this paragraph. Needs better linking and explanation.

Ans: Thanks for the comment, and the revised version will focus on discussing the potential impacts of social character and pre- and post- disaster. Therefore, the research question will be stated in the introduction and bounded rationality will be taken away.

18.Line 229 – what do you mean by internal control?

Ans: Thanks for the comment, and the revised version will focus on discussing the potential impacts of social character and pre- and post- disaster. Therefore, the research question will be stated in the introduction and internal control will be taken away.

19.Line 228-231 repeats exactly some sentences in the introduction – reword appropriately

Ans: Thanks for the comment, and we will rewrite section 3 and 4 in the revised version.

20.The discussion needs some more explanation of the limitations – a limitation and future research section would be ideal. They are touched on in the conclusion (e.g., time limitation), but lack enough detail for the reader to evaluate and interpret.

Ans: Thanks for the comment, and the revised version will add up the explanations of the limitation in the conclusion section.

21.The authors introduce the term 'subjective resilience' but need to define and explain this further. How does it relate to the various measures discussed?

Ans: Thanks for the comment. "Subjective resilience" has been discussed in the introduction, but it has less connection with the rest of discussion. Therefore, we will reevaluate whether use risk perception and adaptive behavior only in the revised version.

22.The authors refer to 'sex' (see section 3.1 in particular), when I believe they need to be referring to 'gender' as the issues here relate to familial responsibilities and social roles relating to someone's gender – not their biological sex. See Rushton et al (2019) for more.

Ans: Thanks for the comment. According to the previous articles, both sex and gender could be applied for such category. However, gender indeed might be more appropriate and we will revise it.

23.Figures need improving for clarity. Figure 1a-d need to be larger as the keys are hard to read, Figure 2 would be better in an appendix. All figures would benefit from

more explanatory extensive captions that enable them to be read and interpreted more easily. Figures 3, 4, 5, 6 & 7 are not in a format I am familiar with. It took a while to interpret them, I feel they need some much clearer captions and further explanation to facilitate interpretation.

Ans: Thanks for the comment, and we will amplify Figure 1a, and take other reviewers' comment we will delete 1b-d for the figures could only reveal limited information. Figures 3 to 7 are radical new here, and we will take the comments seriously and bring up tables or readable figures in the later version.

24.Some of the tables are a bit unclear, e.g., Table 2: it is hard to follow which of the rows in column 1 apply to which rows in the other columns. Can they be reformatted to aid comprehension?

Ans: Thanks for the comment, and we will have revised the table in order to increase readability.

25. References ======== Rushton A, Gray L, Canty J, & Blanchard K (2019) Beyond binary: (re)defining "gender" for 21st century disaster risk reduction research, policy, and practice Int J Environ Res Public Health, https://doi.org/10.3390/ijerph16203984. McClure J, Allen MW, & Walkey F (2001) Countering fatalism: Causal information in news reports affects judgments about earthquake damage Basic Appl Soc Psych, https://doi.org/10.1207/S15324834BASP2302_3. Paton D, Smith L, & Johnston D (2005) When good intentions turn badâËŸA'r: promoting natural hazard preparedness Aust J Emerg Manag 20(1) 25–30. Wei H-L, & Lindell MK (2017) Washington households' expected responses to lahar threat from Mt. Rainier Int J Disaster Risk Reduct 22 77–94, https://doi.org/10.1016/J.IJDRR.2016.10.014. Senkbeil JC, Scott DA, Guinazu-Walker P, & Rockman MS (2014) Ethnic and Racial Differences in Tornado Hazard Perception, Preparedness, and Shelter Lead Time in Tuscaloosa Prof Geogr 66(4) 610–620, https://doi.org/10.1080/00330124.2013.826562. Blake D, Marlowe J, & Johnston D (2017) Get prepared: Discourse for the privileged? Int J Disaster Risk Reduct 25 283–288, https://doi.org/10.1016/J.IJDRR.2017.09.012. Olofsson A, & Rashid S (2011) The White (Male) Effect and Risk Perception: Can Equality Make a Difference? Risk Anal 31(6) 1016–1032, https://doi.org/10.1111/j.1539-6924.2010.01566.x. Lindell MK, & Prater C (2002) Risk area resident' perceptions and adoption of seismic hazard adjustments J Appl Soc Psychol 32(11) 2377–2392. Palmer CGS (2003) Risk perception: Another look at the "white male" effect Heal Risk Soc 5(1) 71–83, https://doi.org/10.1080/1369857031000066014. Dooley D, Catalano R, Mishra S, & Serxner S (1992) Earthquake Preparedness: Predictors in a Community Survey J Appl Soc Psychol 22(6) 451–470, https://doi.org/10.1111/j.1559-1816.1992.tb00984.x. Bateman JM, & Edwards B (2002) Gender and Evacuation: A Closer Look at Why Women Are More Likely to Evacuate for Hurricanes Nat Hazards Rev 3(3) 107–117, https://doi.org/10.1061/(ASCE)1527-6988(2002)3:3(107). Lindell MK, & Perry RW (2012) The Protective Action Decision Model: Theoretical Modifications and Additional Evidence Risk Anal 32(4) 616–632, https://doi.org/10.1111/j.1539-6924.2011.01647.x. Becker JS, Paton D, & Johnston DM (2014) Societal Influences on Earthquake Information Meaning-Making and Household Preparedness Int J Mass Emerg Disasters 32(2) 317–352. Lindell MK, & Hwang SN (2008) Households ' Perceived Personal Risk and Responses in a Multihazard Environment , https://doi.org/10.1111/j.1539-6924.2008.01032.x. Lindell MK, & Prater C (2002) Risk area resident' perceptions and adoption of seismic hazard adjustments J Appl Soc Psychol 32(11) 2377–2392. Becker JS, Paton D, Johnston DM, Ronan KR, & McClure J (2017) The role of prior experience in informing and motivating earthquake preparedness Int J Disaster Risk Reduct 22 179–193, https://doi.org/10.1016/J.IJDRR.2017.03.006. McClure J, Henrich L, Johnston DM, & Doyle EEH (2016) Are two earthquakes better than one? How earthquakes in two different regions affect risk judgments and preparation in three locations Int J Disaster Risk Reduct 16 192–199, https://doi.org/10.1016/j.ijdrr.2016.03.003. Paton D, Anderson E, Becker J, & Petersen J (2015) Developing a comprehensive model of hazard preparedness: Lessons from the Christchurch earthquake Int J Disaster Risk Reduct 14 37–45, https://doi.org/10.1016/j.ijdrr.2014.11.011. Doyle EEH, McClure J,

Potter SH, Becker JS, Johnston DM, Lindell MK, Johal S, Fraser SA, & Coomer MA (2018) Motivations to prepare after the 2013 Cook Strait Earthquake, N.Z. Int J Disaster Risk Reduct 31 637–649, https://doi.org/10.1016/j.ijdrr.2018.07.008.

Ans: Thanks for the long list for the references. We will include as many as we can according to our research question.

---

## Author Response (AR1)

Response to Reviewer 1

As the title indicates, the objective of this manuscript is to explain changes in seismic risk perception and adaptation behavior after an earthquake among different demographic groups. The literature review references a number of relevant citations but also cites tangentially related and outdated citations and overlooks two important reviews and some very relevant recent citations (see the list below). The Introduction fails to state specific research questions or research hypotheses. The data set appears to be excellent but the procedures for sampling cases and measuring items are inadequately described. The Results section is subdivided by the major demographic variables, but those headings don't accurately describe the presentation of results some of which are about pretest-posttest differences that appear to be unrelated to the demographic variables. Moreover, the results are presented in a series of unconventional figures that fail to provide the reader with adequate information about the effects sizes for the impact of the demographic variables on the dependent variables or correlations among dependent variables. The Discussion and Conclusions focus on the effects of the demographic variables on risk perception and adaptation behavior but ignore the pretest-posttest differences. This is a significant limitation because these sections fail to address a major part of the study's stated objective. In addition, systematic reviews of the disaster research literature indicate that demographic variables have small and inconsistent effects on adaptation behavior, so the authors are probably focusing on the least important part of their study's results. Finally, as a general comment, I know from personal experience how difficult it is to submit papers that is not written in my native language. Accordingly, I seek the assistance of a professional editor before submitting papers in other languages. The authors of this manuscript should have done this already and should definitely do so before resubmission.

Ans: Thank you for the general and specific comments, which have been very helpful in improving the research. Indeed, English is not our native language; thank you very much for the recommendation. In fact, this paper has been submitted for English proofreading before submitting to Natural Hazards and Earth System Sciences. We have transferred these valuable comments to American Journal Experts, and the resubmission will be re-edited again by native English speakers.

The following are the point-by-point responses.

1. Line Comment 44 The section on risk perception cites literature that is either overly general (Eagly and Chaiken, 1993, is about attitudes rather than risk perception) or outdated (Sjöberg, 2000; Sjöberg, 1996). Moreover, although risk perception might be influenced by internal and external factors, it does not "sum up" those factors.

Ans: Thank you for the comment. The purpose of this article is to explore the change in risk perceptions and adaptation behaviors between the pre- and postearthquake periods. To identify the main research topic, the revised version has improved both risk perceptions and the potential influence of disaster experience based on the comments (please see lines 35-59).

"It is necessary to minimize disaster risk and build resilience by self-evaluating the capabilities and capacities in responding to risk, that is, preparedness (Jones and Tanner 2017). Being prepared for a future disaster requires various components, such as sufficient personal character, social connections, and financial affordability (Baker and Cormier, 2015). People who are included in vulnerable minority groups and marginalized people might not be able to prepare in advance (Blake et al., 2017). Therefore, an increasing number of studies have emphasized measuring risk perceptions at the individual and household levels (Brown and Westaway 2011; Adger et al. 2009). The perception of disaster risk does not represent a direct function of the probability that threatening events will occur; rather, risk perception captures many other factors, such as attitude, cognition, the degree of danger comprehension, and vulnerability (Sjöberg 2000; Sjöberg 1996; Eagly and Chaiken 1993). Despite the substantial literature illustrating the origin (Barrows, 1923), concept (Sjöberg 2000; Sjöberg 1996), formation (Lindell et al., 2016; Whitney et al., 2004; Wu and Lindell, 2004; Lindell and Perry, 2000), and physical and social contexts of disaster risk perceptions (Blanchard-Boehm and Cook, 2004; Peacock et al., 2005; Peacock, 2003), less attention has been paid to systematically examining changes in risk perceptions.

In fact, disaster experiences might facilitate or constrain preparedness (Becker et al., 2017; Ejeta et al., 2015; Lindell and Perry, 2011; Bostrom, 2008), and such effects might be biased across disasters, cultures or regions. A disaster resulting in limited impacts or the assumption that a future disaster will not occur might encourage people to not prepare for future disasters (Paton et al., 2014; Barron and Leider, 2010). Alternatively, people might take any adaptation approaches based upon damage or losses, physical injury, emotional injury and so on (Perry and Lindell, 2008; Nguyen et al., 2006; Heller et al., 2005). The physical damage or losses (Solberg et al., 2010) and psychological fear or anxiety (Rüstemli and Karanci, 1999) resulting from disaster experiences could motivate adaptation behaviors. However, socioeconomic characteristics such as income, age, and gender might encourage or discourage individuals from taking adaptive actions (Bankoff 2006; Wisner et al. 2004). For example, if people

cannot act adequately to mitigate such anxiety, they might take no actions at all (Paton and McClure, 2013). Due to limited knowledge and resources, people tend not to respond to common disasters and tend to have personal preferences for disasters, such as denying disasters, denying disaster probability, and having certain beliefs about the government and public infrastructure. Therefore, examining risk perceptions and adaptation behaviors based on various socioeconomic characteristics could provide important information for disaster management."

2. 64 The title makes it reasonably clear what are the study's research objectives, but there is no clear statement of research questions or research hypotheses at the conclusion of the Introduction. This might be why the Results and Discussion sections fail to adequately describe the changes in risk perception and adaptation behavior.

Ans: Thank you for the insightful comment. Overall, this study contributes to explorations of how earthquake disasters influence the risk perceptions and adaptation behaviors of residents in Taiwan, and it further categorizes them according to their socioeconomic characteristics. Based on past studies, the interactions of socioeconomic characteristics can collectively affect responses to disasters. Therefore, this study discusses such responses based on various socioeconomic characteristics to explore how they affect pre- and postrisk perceptions and adaptation behaviors. The revised version has improved the statement of the research questions in the Introduction and further improved the consistency between the title and the article (please see lines 72-75).

"Based on past studies, the interactions of socioeconomic characteristics can collectively affect responses to disasters. This study discusses such responses based on various socioeconomic characteristics to explore how such characteristics affect pre- and postearthquake risk perceptions and adaptation behaviors."

3. 78 Figure 1a is sufficient for a research article. Figure 1b, 1c, and 1d are only of interest to local authorities.

Ans: Thank you for the comment. To leave accurate information, the revised version has deleted the remaining figures in Figure 1 according to the comment.

4. 89 It is unclear what it meant by "simple random sampling". Is this simple random sampling from a sample frame (i.e., a list of telephone numbers) or random digit dialing?

Ans: Thank you for the comment. To reflect the characteristics of the larger groups, stratified random sampling is employed to determine appropriate sample numbers in 43

smallest-level administrative units. All surveys conducted involved voluntary response sampling. The preearthquake survey is a street survey, and the postearthquake survey is a telephone survey based on phone number databases within the study area conducted by the survey research center of a domestic academic institution. The telephone survey employed a computer-assisted telephone interview (CATI) system. The interviewers followed a script provided by a software application with higher quality assurance monitoring.

5.  99 The section describing the measures should not be referring to the research literature. Those references should have already been cited in the Introduction's literature review. Instead, this section should specifically describe each item in the questionnaire and how it was measured. Thus, the description of the items "probability of an earthquake disaster occurring within ten years", "fear of earthquake", and "worry of building collapse" should list the exact English translation of those items and list the rating scale anchors that were used (e.g., "Not at all = 1 to Almost a certainty = 7" for the earthquake probability rating). The items measuring "the impacts they expected from the disaster" should be replaced by a statement of the specific impacts that were listed.

Ans: Thank you for the comment. The purpose of section 2.3 is to illustrate the survey items in the study. To separate the data and literature review, the updated version has revised this section and focused on explaining the variables used. In addition, the revised version adds a new Table 1 to explain the measurement of the questionnaires (please see lines 109-126 and Table 1).

"Perceived risk is not necessarily equivalent to the probability of occurrence of a disaster. Rather, it summarizes many other factors. Increasing research focuses on the risk perceptions of earthquake disasters, and such perceptions might vary. Previous studies have shown that terror often accompanies changes in the physical environment, the loss of human lives and the destruction of property. Therefore, among earthquake-related stressors, we were concerned with individuals' perceptions of the probability of an earthquake disaster occurring within ten years and the impacts they expected from such a disaster, including fear of earthquakes and worries over buildings collapsing.

Although prior disaster experiences and observation of the natural environment might form disaster perceptions, various socioeconomic characteristics might further affect such perceptions. Adaptation behavior is a way for individuals to adapt their living environment to new events that may occur and impact the existing system. People who have faith in adaptation behaviors might take whatever approaches they have, while others might take no such approaches. Therefore, in the adaptation behavior section, we were concerned with the ways in which people respond to earthquake disasters. To survive earthquakes, seismic restraints might play important roles during such disasters.

Hence, there are two items regarding house retrofitting, including the willingness to retrofit houses and house retrofitting after professional assessment.

There are five items in the survey to explore both risk perceptions and adaptation behaviors. Risk perceptions are measured by three items on the expected impacts of earthquakes, and adaptation behaviors are measured by two items on the willingness to support policies. The measurement, shown in Table 1, combines 7-point Likert-scale items and Yes/No questions (see Table 1). A transformation process is conducted to solve the problems posed by scales with different measurement systems."

**Table 1** Measurement of the questionnaires.

| Aspects | Items | predisaster | postdisaster |
|---|---|---|---|
| Risk perceptions | Probability of an earthquake disaster occurring within the next ten years | 7-point | 7-point |
| | Fear of earthquakes | 7-point | 7-point |
| | Worries over buildings collapsing | 7-point | 7-point |
| Adaptation behaviors | Willingness to retrofit houses | Yes/No | 7-point |
| | Willingness to retrofit houses after assessment | Yes/No | 7-point |

Completely disagree = 1 to completely agree = 7

6. 114 Most of the first paragraph in this section is, or should be, common knowledge among survey researchers. Consequently, all but the last sentence should be deleted as should Figure 2.

Ans: Thank you for the comment. The first paragraph in section 2.4 aims to give a general concept of ANOVA to readers. However, it is indeed common knowledge among survey researchers. Therefore, the revised version has deleted the first sentence because it is too general, but it keeps the second sentence regarding one-way analysis of variance. The revised version has kept Figure 2 to let the readers grasp the overall procedure of ANOVA. (Please see lines 128-145)

"One-way ANOVA is an extension of the independent samples t-test that can be used to compare any number of groups (Bewick et al. 2004; Whitely and Ball 2002). The core value of one-way ANOVA lies in the ability to examine means that are significantly different from each other between groups. One-way ANOVA is calculated as follows:

$$\frac{\sum_{i=1}^{n}(x_i - \bar{x})^2}{n-1} \tag{1}$$

where the variance comes from a set of n values $(x_1, x_2, ..., x_n)$ and the degrees of freedom is n-1. In one-way ANOVA, the F statistic test is used and represented equally among groups. A significant F statistic test result indicates a significant difference between groups, and the P-value of 0.05 is the common threshold. First, Levene's test is applied to examine the null hypothesis that the variance is equal across groups. A result of Levene's test lower than 0.05 indicates that it is necessary to apply Welch's test because there is no equal variance between groups. On the other hand, if the result of Levene's test is greater than 0.05, then we can depend on the ANOVA results. Overall, a significant F statistic in both Welch's test and ANOVA indicates that at least two groups are different, but it does not identify which groups are different from the others. However, a P-value lower than 0.05 indicates significance or the probability of a type II error, which is the possibility of incorrectly rejecting the null hypothesis or wrongly concluding a difference between groups. Therefore, a post hoc test and multicomparison analysis testing are necessary to avoid type II errors and to further examine the differences between levels. Due to the assumption of homogeneity of variance, we then apply the Games-Howell test and Benjamini-Hochberg procedure.

Quantitative data analysis was conducted using the Statistical Package for Social Scientists (SPSS) software. Each response to the items in the questionnaire survey was rated on a scale ranging from 1 to 7, with 1 as the highest level of vulnerability (or lowest level of resilience) and 7 as the lowest level of vulnerability (highest level of resilience)."

7.    142 Table 1 should also contain data for the distributions of gender, age, education, occupation, and homeownership for the study area so readers can assess the extent of sample bias.

Ans: Thank you for the comment. The table could be further improved to present the distributions of both the sample and the study area. Therefore, the revised version has added relevant data for readers to assess the extent of sample bias. (Please see Table 2)

"**Table 2** Sample characteristics in the pre- and postearthquake surveys.

| Characteristics | Pre- | Post- | Study area | Characteristics | Pre- | Post- | Study area |
|---|---|---|---|---|---|---|---|
| Gender | | | | Occupation* | | | |
| Male | 53.38% | 44.89% | 49.27% | Students | 9.09% | 7.23% | 38.53% |
| Female | 46.42% | 55.11% | 50.73% | Home Makers | 10.96% | 18.94% | |

| Characteristics | Pre- | Post- | Study area | Characteristics | Pre- | Post- | Study area |
|---|---|---|---|---|---|---|---|
| **Age** | | | | White-collar Workers | 37.76% | 32.55% | 59.08% |
| < 15 yr. | 7.46% | 1.70% | 13.97% | Blue-collar Workers | 41.96% | 41.28% | |
| 15-40 yr. | 38.23% | 28.30% | 37.96% | **House Ownership*** | | | |
| 40-60 yr. | 37.53% | 51.91% | 32.16% | Self-owned | 48.95% | 63.62% | 85.93% |
| > 60 yr. | 16.78% | 18.09% | 15.91% | Family-owned | 32.17% | 32.34% | 3.20% |
| **Education** | | | | Renting | 18.65% | 4.04% | 7.82% |
| Elementary/Junior High | 21.68% | 21.91% | 21.63% | | | | |
| High School | 47.32% | 41.49% | 30.54% | | | | |
| University/Graduate | 31.00% | 36.60% | 46.96% | | | | |

Note 1: The values without official statistics are replaced by data from the Tainan Municipality.
Note 2: The share of illiterate individuals in the study area is 0.87%.
Note 3: The official statistics for occupation are categorized into employment and unemployment, and the unemployment percentage is 2.39%. In addition, neither students nor home makers are included in the labor force.
Note 4: The official statistics for house ownership include self-owned, family-owned, renting, and others, and the percentages are 85.93%, 3.20%, 7.82%, and 3.05%, respectively.."

8. 144 Section 3.1 is labeled sex but presents a number of results that appear to be unrelated to sex differences. Specifically, "the earthquake probability (the P value of 0.049), the fear =of earthquake (the P value of 0.000), and the willingness on house retrofit (the P value of 0.002) are statistical significance indicating a serious earthquake indeed increase awareness of disaster" seems to be a pretest-posttest comparison that is unrelated to sex differences. This problem continues throughout the rest of the Results section.

Ans: Thank you for the valuable comment. The purpose of this article is to explore the changes in risk perceptions and adaptation behaviors based on various socioeconomic characteristics between the pre- and posteearthquake periods. Therefore, the revised version has emphasized this discussion in the results section (please see lines 148-218)

[revised manuscript text omitted]

9.  148 Figure 3 presents the results in a format that is rather inventive, but extremely confusing and relatively uninformative, compared to the conventional method of presenting a matrix containing the variables' means in the first column, the standard deviations in the second column, and the intercorrelations in the remaining columns. In addition providing effect sizes for to the impact of the independent variables on the dependent variables, a correlation matrix allows the reader to see the correlations among the dependent variables (see Lindell & Hwang, 2008, for an example). Providing this correlation matrix will eliminate the need for Figures 4-7, as well.

Ans: Thank you for the comment. Because the purpose is to compare changes over time based on various socioeconomic characteristics, the arrows and the lines are used to express such outcomes. However, as mentioned by the reviewer, the figure might not be a perfect way to present the results and make them more confusing. Therefore, conventional tables are applied to show the overall results for the socioeconomic characteristics (please see Table 3 to Table 10).

10. 191 The Discussion section only addresses the effects of the demographic variables, ignoring the effects of changes in risk perception and their possible effects on risk reduction actions.

Ans: Thank you for the comment. Based on past studies, the interactions of socioeconomic characteristics can collectively affect responses to disasters. Therefore, the purpose of this article is to discuss such responses based on various socioeconomic characteristics to explore how they affect pre- and postrisk perceptions and adaptation behaviors. The revised version has rewritten the discussion section on the potential impacts of the interactions of socioeconomic characteristics on changes in disaster perceptions and adaptation behaviors (please see lines 220-252).

"4 Discussion

According to the results, after the Meinong earthquake, people tended to have greater risk perceptions regarding future earthquakes but were less willing to retrofit their houses. The findings show that people might become less willing to prepare, which is quite similar to the result of a survey conducted after the 2011 Christchurch earthquake (Statistics New Zealand, 2012; Paton and Johnston, 2008). In fact, the relationship between disaster experience and preparedness has been regarded as a key issue based on the recommendations of the Sendai Framework (United Nations, 2015). According to past studies, it is difficult for people to imagine any consequences if they lack earthquake experience (Paton and McClure, 2013). However, the study finds that the levels of disaster preparedness become low after serious disasters. Therefore, disaster experience might not necessarily increase people's willingness to prepare. On the other hand, socioeconomic characteristics might still affect the decision-making process with regard to adopting adaptation behaviors.

In terms of gender, females show greater fear and worries regarding future earthquake disasters than males, while they have a similar willingness to retrofit their houses (see Fig. 3). According to past studies, the responses of women might be more internal and backstage, whereas those of men might be more external and front stage (Enarson 2001; Always et al. 1998; Fordham 1998). The economic status and family role of women might forbid possible adaptive choices compared to men (Tobin-Gurley and Enarson 2013). Men, in contrast, are more risk tolerant than women (Finucane et al. 2000). Although gender inequality prevails in different ways around the world, women's safety concerns for their family have been well documented in both environmental protection movements and neighborhood emergency preparedness campaigns (Litt et al. 2012; Luft 2008; Erikson 1994; Turner et al. 1986). Therefore, it is necessary to provide more diverse options for house retrofitting for families to increase their potential willingness to improve the anti-seismic resilience of their houses.

Regarding education, people tend to become aware of earthquake risk after a serious disaster event, and there are no significant variations between educational level categories. Although there is a significant decrement in the result for house retrofitting, people who have a higher level of education might be more willing to retrofit their houses (see Fig. 4). There are similarities in occupation; people who are white-collar workers are still much more willing to retrofit their houses than blue-collar workers, home makers, and students. In addition, home makers have higher risk perceptions than those belonging to the other occupation categories. Available resources might be the key factor affecting whether people prepare for and respond to disasters. Social stratification plays a role in perceiving and reacting to risk, including people's understanding of disaster information, the sources announcing disaster information, and potential options to respond (Fothergill and Peek 2004).

Gender, age, and class alone do not make people vulnerable, while the interactions between factors might result in an increase in vulnerability. Overall, social characteristics do indeed affect decisions regarding disaster awareness and adaptation behaviors. In addition, disaster experience does indeed facilitate local awareness but constrains preparedness in regard to Taiwan's earthquake experience. Among gender, education, and occupation, each category shows a similar tendency of increased risk awareness of risk but decreased willingness to retrofit houses. However, over time, risk awareness might

fade away. Therefore, risk communication, risk education, and diverse mitigation options are required as soon as possible after serious earthquakes to help people be ready for future events."

Response to Reviewer 2

The piece of the change of risk perception and adaptation behavior between pre and post-earthquake disaster proposes an interesting comparative discussion. The manuscript has a clear scope but some sections could be improved. In addition, there are some other literature exploring similar topics (listed below) and should be included in the discussion. Indeed, risk perception and adaptive actions might be varied according to different social characters. The presentation of result is radical different from previous studies in ANOVA. Traditional table could reveal various value and significance. Authors should provide more information of such different expression to let reader catch such outcome. As a whole, the dataset is interesting and meaningful for most studies indeed could only examine pre- or post- earthquake only.

Ans: Thank you for the general and specific comments, which have been very helpful in improving the research. First, thank you for providing related references for this article; the revised version includes certain works. It seems that the current presentation of the results might confuse readers, and the revised version takes the comments into account to alleviate such confusion.

In the following, I would like to separate my comments into general and specific.

1. Although risk perception and adaptation behavior are the key issue, it seems that disaster experience is the key factor authors discussed in this article. The overall logic in introduction is blurred right now, and such vague might further the results interpretation. How to reconnect the research question and the findings might be important for this study.

Ans: Thank you for the comment. The study attempts to discuss changes in risk perceptions and adaptation behaviors based on various socioeconomic characteristics between pre- and postearthquake disaster periods. The research question is not clear enough in the current version, and the revised version improved such statements in both the "Introduction" and "Conclusions." The clear research question might help to reconnect the motivation and findings (Please see lines 23-79).

"1 Introduction

[revised manuscript text omitted]

In summary, the threats in a given area posed by future earthquakes with a magnitude larger than that experienced in the past create uncertainty in regard to the ability to mitigate impacts to acceptable levels using only engineering or construction measures. Humans have the capacity to respond to the environment to reduce risk by learning from past experience, and changes in attitudes and behaviors are very helpful in responding to earthquake disasters (Gifford 2014). Theoretically, a more accurate measurement and tracking of the interactions of socioeconomic characteristics that collectively affect responses to disasters might help support the right activities and target the right people in disaster management (Oddsdottir et al. 2013; Adger 2000). Past studies have placed more emphasis on predisaster conditions to explore the interactions of individuals' decisions (Levine 2014). Examining predisaster and postdisaster conditions could reveal the impact of extreme events and how people's perceptions of such events and their willingness to take potential adaptation approaches might change. Therefore, this study contributes by exploring how earthquake disasters influence the risk perceptions and adaptation behaviors of residents in Taiwan and further categorizes them according to socioeconomic characteristics. The sample is of particular interest because it contains pre- and postdisaster information on residents who were directly affected by the Meinong earthquake (participants completed surveys approximately 1 year before and 3 months after the earthquake), allowing a more robust analysis of the effects of natural disasters on subjective resilience compared to previous research. Based on past studies, the interactions of socioeconomic characteristics can collectively affect responses to disasters. This study discusses such responses based on various socioeconomic characteristics to explore how such characteristics affect pre- and postearthquake risk perceptions and adaptation behaviors. In addition to the introduction, this paper is organized as follows. Section 2 provides a brief description of the research design, including the study area, the data collection, the measures for subjective resilience, and the methods. Section 3 presents the comparative analysis between pre- and postdisaster surveys based on the results of one-way analysis of variance (ANOVA). Section 4 presents the comparative analysis between our findings and those of past studies. The final section offers some conclusions."

2. The expression for the results need more information. It is easy for readers to catch the results from table such as the value of the variable and the p-value. Although in Figure 3 to 7 there is a red line for p-value of 0.05, the figures are still blurred. What does the arrow mean? In order to increase readability, certain information might be necessary to provide.

Ans: Thank you for the comment. The arrows in Figures 3 to 7 indicate the changes in disaster perceptions and adaptive behaviors. The current presentation is confusing, and the revised version presents the findings based on traditional ANOVA to clarify the results (please see Table 3 to Table 10)s

3.   Line 35. Current reference applied to risk perception and adaptation behavior is rather too old. In fact, there are more recent literature exploring similar issues or topics. Although some of the literature are important such as Lindell, Becker, Sjöberg and so on, it is important to update such discussion.
Motivations to prepare after the 2013 Cook Strait Earthquake, N.Z Perceptions and reactions to tornado warning polygons: Would a gradient polygon be useful? Assessment of households' responses to the tsunami threat: A comparative study of Japan and New Zealand Perceptions, behavioral expectations, and implementation timing for response actions in a hurricane emergency Port stakeholder perceptions of Sandy impacts: a case study of Red Hook, New York Conflicts in adaptation: case studies from Nepal and the Maldives The role of prior experience in informing and motivating earthquake preparedness

Ans: Thank you for the comment. The revised version has taken the suggested references into consideration and improved the relevant statements (please see lines 35-59)

"It is necessary to minimize disaster risk and build resilience by self-evaluating the capabilities and capacities in responding to risk, that is, preparedness (Jones and Tanner 2017). Being prepared for a future disaster requires various components, such as sufficient personal character, social connections, and financial affordability (Baker and Cormier, 2015). People who are included in vulnerable minority groups and marginalized people might not be able to prepare in advance (Blake et al., 2017). Therefore, an increasing number of studies have emphasized measuring risk perceptions at the individual and household levels (Brown and Westaway 2011; Adger et al. 2009). The perception of disaster risk does not represent a direct function of the probability that threatening events will occur; rather, risk perception captures many other factors, such as attitude, cognition, the degree of danger comprehension, and vulnerability (Sjöberg 2000; Sjöberg 1996; Eagly and Chaiken 1993). Despite the substantial literature illustrating the origin (Barrows, 1923), concept (Sjöberg 2000; Sjöberg 1996), formation (Lindell et al., 2016; Whitney et al., 2004; Wu and Lindell, 2004; Lindell and Perry, 2000), and physical and social contexts of disaster risk perceptions (Blanchard-Boehm and Cook, 2004; Peacock et al., 2005; Peacock, 2003), less attention has been paid to systematically examining changes in risk perceptions.

In fact, disaster experiences might facilitate or constrain preparedness (Becker et al., 2017; Ejeta et al., 2015; Lindell and Perry, 2011; Bostrom, 2008), and such effects might be biased across disasters, cultures or regions. A disaster resulting in limited impacts or the assumption that a future disaster will not occur might encourage people to not prepare for future disasters (Paton et al., 2014; Barron and Leider, 2010). Alternatively, people might take any adaptation approaches based upon damage or losses, physical injury, emotional injury and so on (Perry and Lindell, 2008; Nguyen et al., 2006; Heller et al., 2005). The physical damage or losses (Solberg et al., 2010) and psychological fear or anxiety (Rüstemli and Karanci, 1999) resulting from disaster experiences could motivate adaptation behaviors. However,

socioeconomic characteristics such as income, age, and gender might encourage or discourage individuals from taking adaptive actions (Bankoff 2006; Wisner et al. 2004). For example, if people cannot act adequately to mitigate such anxiety, they might take no actions at all (Paton and McClure, 2013). Due to limited knowledge and resources, people tend not to respond to common disasters and tend to have personal preferences for disasters, such as denying disasters, denying disaster probability, and having certain beliefs about the government and public infrastructure. Therefore, examining risk perceptions and adaptation behaviors based on various socioeconomic characteristics could provide important information for disaster management."

4. Line 51. The research question might need more specific and elaborated in the last paragraph of Introduction section. Although the title is rather clear, there is no statement regarding the research question. Therefore, this part could be improved.

Ans: Thank you for the comment. The revised version has added the research questions in both the introduction and conclusions to improve the overall logic in the study (please see lines 72-75).

"Based on past studies, the interactions of socioeconomic characteristics can collectively affect responses to disasters. This study discusses such responses based on various socioeconomic characteristics to explore how such characteristics affect pre- and postearthquake risk perceptions and adaptation behaviors."

5. Line 85. In the article, the survey data is the main dataset. "All survey sampling methods relied on simple random sampling." How can you tell the representative of the sampling data? What is the ratio between sampling amount and the study area?

Ans: Thank you for the comment. To reflect the characteristics of larger groups, stratified random sampling is employed to determine appropriate sample numbers in 43 smallest-level administrative units. All surveys involved voluntary response sampling. The preearthquake survey is a street survey, and the postearthquake is a telephone survey based on phone number databases within the study area conducted by the survey research center of a domestic academic institution. The telephone survey employed a computer-assisted telephone interview (CATI) system. The interviewers followed a script provided by a software application with higher quality assurance monitoring.

6. Figure 2 is important for this study. However, it is unclear which result is applied post hoc or not. This should be discussed systematically either in the research design or in the results.

Ans: Thank you for the comment. The revised manuscript has rewritten both section 3 "results" and section 4 "discussion" (please see lines 147-218).

7.    Line 135. The separation of the result is based upon social character. Again, due to there is no specific research question, it is hard for readers to understand why separate in current sub-categories. In addition, I think pre- and post- is the main concern, and this should be clarified.

Ans: Thank you for the comment. Indeed, the main concerns are socioeconomic characteristics pre- and postearthquake. Therefore, the revised version has rewritten section 3 and section 4 (please see lines 147-218).

[revised manuscript text omitted]

8.  Line 191. Figure 8 to 10 are not providing enough information for the readers, so I suggest these figure could be deleted.

Ans: Thank you for the comment. Originally, the purpose of Figures 8 to 10 was for future disaster management by taking into account socioeconomic characteristics. However, the figures might not have provided enough information; therefore, the revised version deleted them.

Response to Reviewer 3

This is an interesting study investigating risk perception and preparedness actions through surveys, pre and post an earthquake event. The findings are important and contribute to the growing body of research in this space. However, currently this manuscript requires significant revisions for those findings to be recognised clearly. Primarily, areas of research are missing in the introduction and discussion, that both can provide more context and help the authors interpret some of their findings. Second, there is a lack of clarity in a number of places, including the presentation of methodology, results, and figures. Thirdly, while I appreciate the challenge of writing in a second language (and acknowledge the privilege of being able to write in my first), the manuscript is currently very difficult to read and understand in places – which sadly detracts from the data within. I would recommend the editors consult a professional editing service, seek assistance from the journal if it offers that option, or seek an additional author to assist with the writing.

Ans: Thank you for the valuable and insightful comments. First, thank you for pointing out that there are areas of research missing in the introduction and discussion; the revised version focuses on these two sections to improve both statements and interpretation. Second, the submitted paper has undergone English proofreading by American Journal Experts (AJE), and the comments have been transferred to AJE. Hopefully, the revised version resolves such issues.

My detailed recommendations follow below,

1.  Some substantial areas of research are missing, including key preparedness and response literature. For example, discussion of the Protective Action Decision Model (Lindell & Perry, 2012; Lindell & Hwang 2008; Lindell & Prater 2002) and other hazard preparedness models (e.g., Paton et al., 2015), societal influences on household preparedness (Becker et al 2014), recent research on preparedness motivations (Becker et al 2017; Doyle et al 2018) which consider multiple-event risk perceptions (McClure et al 2016, Doyle et al 2018). Thus, the introduction and contextual discussion of the findings should explore some of the elements raised in this pre-existing literature, and how the results relate to those, including self- and collective-efficacy, outcome expectancy, responsibility, etc. Notably, the discussion omits a number of key texts investigating the relationship between gender and preparedness, and reasons for that difference (e.g., familial responsibilities), e.g. Dooley et al; Bateman and Edwards; Lindell & Prater; Olofsson & Rashid; Palmer; as well as texts exploring the barriers to

preparedness (e.g., Blake et al 2018; Senkbeil et al 2014). The Literature on attribution theory, and on trust in communications is also lacking.

Ans: Thank you for the valuable comments and for providing abundant references for us. Based on past studies, the interactions of socioeconomic characteristics can collectively affect responses to disasters. This study discusses such responses based on various socioeconomic characteristics to explore how they affect pre- and postdisaster risk perceptions and adaptation behaviors. The revised version has taken the suggested areas of research into consideration and improved the statements in both the introduction and discussion sections (please see lines 35-59 and 219-252).

[revised manuscript text omitted]

2. The authors need to set the scene more in section 2.1 (Study Area) – what resilience building activities have been conducted in these regions, if at all? Some more information about the community, and previous events or resilience activities is needed.

Ans: Thank you for the comment. The revised version includes more information related to resilience achievements regarding the study area in the first paragraph of section 2.1.

3. Section 2.2 – 'simple random sampling' – of what? The phone records?

Ans: Thank you for the comment. To reflect the characteristics of larger groups, stratified random sampling is employed to determine appropriate sample numbers in 43 smallest-level administrative units. All surveys involved voluntary response sampling. The preearthquake survey is a street survey, and the postearthquake is a telephone survey based on phone number databases within the study area conducted by the survey research center of a domestic academic institution. The telephone survey employed a computer-assisted telephone interview (CATI) system. The interviewers followed a script provided by a software application with higher quality assurance monitoring.

Forty-three

The pre-disaster survey relied on stratified random sampling, and sample number in 43-neighborhood unit is based to the population size.

The revised version will improves the illustration of data collection.

4. Section 2.2 – what do you mean by 'some notifications'?

Ans: Here, "some notification" indicates giving some information to the respondents, such as when the last time there was an earthquake with a magnitude over 6.0 was, where the fault line is, what the frequency of earthquake disasters in the study area is. The current version did not state this clearly, and the revised version improves the

statement in data collection description (please see lines 102-104).

"The respondents were reminded of some particular information regarding the most recent earthquake, the geographic location of the nearest fault line, the impact of the disaster event, the frequency of earthquakes in the study area, etc. Additionally, the scale of earthquake magnitude is defined as over 6.0."

5.  Section 2.2 – for the survey questions, why did you choose these particular factors (e.g., trust in government and responsibility attribution?). These factors need more detail explanation in the introduction, referencing the relevant literature (e.g., on trust and attribution theory for risk communication), such that in section 2.2 there is more rationale and explanation for their choice (and prioritisation over other potential factors).

Ans: Thank you for the comment. This study has reviewed past research on the pre- and postdisaster impacts of socioeconomic characteristics. Our purpose was to apply items to detect individuals' disaster perceptions and adaptation behaviors. The revised version has explained the reasons and principles of the factors applied in this article (please see lines 110-127)

"Perceived risk is not necessarily equivalent to the probability of occurrence of a disaster. Rather, it summarizes many other factors. Increasing research focuses on the risk perceptions of earthquake disasters, and such perceptions might vary. Previous studies have shown that terror often accompanies changes in the physical environment, the loss of human lives and the destruction of property. Therefore, among earthquake-related stressors, we were concerned with individuals' perceptions of the probability of an earthquake disaster occurring within ten years and the impacts they expected from such a disaster, including fear of earthquakes and worries over buildings collapsing.

Although prior disaster experiences and observation of the natural environment might form disaster perceptions, various socioeconomic characteristics might further affect such perceptions. Adaptation behavior is a way for individuals to adapt their living environment to new events that may occur and impact the existing system. People who have faith in adaptation behaviors might take whatever approaches they have, while others might take no such approaches. Therefore, in the adaptation behavior section, we were concerned with the ways in which people respond to earthquake disasters. To survive earthquakes, seismic restraints might play important roles during such disasters. Hence, there are two items regarding house retrofitting, including the willingness to retrofit houses and house retrofitting after professional assessment.

There are five items in the survey to explore both risk perceptions and adaptation behaviors. Risk perceptions are measured by three items on the expected impacts of earthquakes, and adaptation behaviors are measured by two items on the willingness to support policies. The measurement, shown in Table 1, combines 7-point Likert-scale items and Yes/No questions (see Table 1). A transformation

process is conducted to solve the problems posed by scales with different measurement systems."

6. I do not think the authors need to include the full description of the ANOVA in section 2.4, given it is a well know statistical test. Would recommend trimming, or if needed including in an appendix. Much of section 2.4 can be summarised much more briefly, as these are standard approaches.

Ans: Thank you for the comment. The revised version has trimmed section 2.4 because ANOVA is indeed a well-known statistical test (please see lines 128-147)

**"2.4 Methods: One-way analysis of variance**

One-way ANOVA is an extension of the independent samples t-test that can be used to compare any number of groups (Bewick et al. 2004; Whitely and Ball 2002). The core value of one-way ANOVA lies in the ability to examine means that are significantly different from each other between groups. One-way ANOVA is calculated as follows:

$$\frac{\sum_{i=1}^{n}(x_i - \bar{x})^2}{n-1} \tag{1}$$

where the variance comes from a set of n values $(x_1, x_2, \ldots, x_n)$ and the degrees of freedom is n-1.

In one-way ANOVA, the F statistic test is used and represented equally among groups. A significant F statistic test result indicates a significant difference between groups, and the P-value of 0.05 is the common threshold. First, Levene's test is applied to examine the null hypothesis that the variance is equal across groups. A result of Levene's test lower than 0.05 indicates that it is necessary to apply Welch's test because there is no equal variance between groups. On the other hand, if the result of Levene's test is greater than 0.05, then we can depend on the ANOVA results. Overall, a significant F statistic in both Welch's test and ANOVA indicates that at least two groups are different, but it does not identify which groups are different from the others. However, a P-value lower than 0.05 indicates significance or the probability of a type II error, which is the possibility of incorrectly rejecting the null hypothesis or wrongly concluding a difference between groups. Therefore, a post hoc test and multicomparison analysis testing are necessary to avoid type II errors and to further examine the differences between levels. Due to the assumption of homogeneity of variance, we then apply the Games-Howell test and Benjamini-Hochberg procedure.

Quantitative data analysis was conducted using the Statistical Package for Social Scientists (SPSS) software. Each response to the items in the questionnaire survey was rated on a scale ranging from 1 to 7, with 1 as the highest level of vulnerability (or lowest level of resilience) and 7 as the lowest level of vulnerability (highest level of resilience)."

7. The tale end of section 2.4, outlining the Likert scale of 1 to 7, should be moved

to section 2.3 where the measures are discussed.

Ans: Thank you for the comment. The revised version has moved the Likert scale of 1 to 7 to section 2.3 and added Table 1 "the measurement of questionnaires" (Please see lines 109-127 and Table 1)

"2.3 Measures for risk perceptions and adaptation behaviors

Perceived risk is not necessarily equivalent to the probability of occurrence of a disaster. Rather, it summarizes many other factors. Increasing research focuses on the risk perceptions of earthquake disasters, and such perceptions might vary. Previous studies have shown that terror often accompanies changes in the physical environment, the loss of human lives and the destruction of property. Therefore, among earthquake-related stressors, we were concerned with individuals' perceptions of the probability of an earthquake disaster occurring within ten years and the impacts they expected from such a disaster, including fear of earthquakes and worries over buildings collapsing.

Although prior disaster experiences and observation of the natural environment might form disaster perceptions, various socioeconomic characteristics might further affect such perceptions. Adaptation behavior is a way for individuals to adapt their living environment to new events that may occur and impact the existing system. People who have faith in adaptation behaviors might take whatever approaches they have, while others might take no such approaches. Therefore, in the adaptation behavior section, we were concerned with the ways in which people respond to earthquake disasters. To survive earthquakes, seismic restraints might play important roles during such disasters. Hence, there are two items regarding house retrofitting, including the willingness to retrofit houses and house retrofitting after professional assessment.

There are five items in the survey to explore both risk perceptions and adaptation behaviors. Risk perceptions are measured by three items on the expected impacts of earthquakes, and adaptation behaviors are measured by two items on the willingness to support policies. The measurement, shown in Table 1, combines 7-point Likert-scale items and Yes/No questions (see Table 1). A transformation process is conducted to solve the problems posed by scales with different measurement systems."

"**Table 1** Measurement of the questionnaires.

| Aspects | Items | predisaster | postdisaster |
|---------|-------|-------------|--------------|
| Risk perceptions | Probability of an earthquake disaster occurring within the next ten years | 7-point | 7-point |
| | Fear of earthquakes | 7-point | 7-point |
| | Worries over buildings collapsing | 7-point | 7-point |
| Adaptation behaviors | Willingness to retrofit houses | Yes/No | 7-point |
| | Willingness to retrofit houses after assessment | Yes/No | 7-point |

Completely disagree = 1 to completely agree =7

8. Lines 145 to 150, seems to contradict oneself on first reading – first you say that there is willingness to house retrofit, and then that it decreased. I think you mean pre/post earthquake, but this needs clarification.

Ans: Thank you for the comment. Because the research question is not that clear in the introduction and further results in confusing statements in the findings, the revised version has rewritten the results to make them consistent in stating the pre- and postdisaster changes in the socioeconomic characteristics (please see lines 148-219).

[revised manuscript text omitted]

9.  In general the ANOVA results need clearer reporting, to standard (brief style) formatting including more clearly the F statistic and degrees of freedom, rather than just the P value. To that end, the P value isn't 0.000, but should be reported as p<0.0005.

Ans: Thank you for the comment. The revised version has provided the traditional ANOVA to give clear reports of the results (please see Table 3 to Table 10).

10. Line 140, explain why high school/blue collar might have less capability to

adjust in the introduction, to set the context here.

Ans: Thank you for the comment. The original version has some contradictory statements in the results and discussion. The revised version has rewritten these two sections (please see 148-252).

11. Section 3.2 (Age) is hard to follow, and needs rewording completely, as currently it reads contradictory.

Ans: Thank you for the comment. The original version has some contradictory statements in the results and discussion. The revised version has rewritten these two sections (please see 148-252).

12. Line 190 – this should be explored in the discussion, in the context of how both resources ($) and / or care responsibilities could be a possible interpretation of this finding.

Ans: Thank you for the comment. As we know, households with children could have more willingness to pay more for house retrofitting, and self-owned households could believe in the original house safety or rely on their life experience. However, it is suitable for complete domestic research because of cultural differences and detailed socioeconomic information. That is the reason why we decided to have a comprehensive comparative analysis instead of a specific discussion. We believe it is a good issue for future work.

13. Section 3.3 – does the higher education group correlate with income? Equally, could this group have less worry of buildings collapsing because they could afford to have better buildings to start with (or were able to retrofit)? These nuances need further discussion.

Ans: Thank you for the comment. In this article, we focus on exploring how socioeconomic characteristics pre- and postdisaster risk perceptions and adaptation behaviors. However, there are more topics that could be further discussed, such as the correlation between education and income, occupation and income, and others. It is worth extending this discussion in future works.

14. Line 193 – This discussion of gender needs more explanation in context of the references listed previously.

Ans: Thank you for the comment; the discussion has been rewritten (please see lines 230-238).

"In terms of gender, females show greater fear and worries regarding future earthquake disasters than

males, while they have a similar willingness to retrofit their houses (see Fig. 3). According to past studies, the responses of women might be more internal and backstage, whereas those of men might be more external and front stage (Enarson 2001; Always et al. 1998; Fordham 1998). The economic status and family role of women might forbid possible adaptive choices compared to men (Tobin-Gurley and Enarson 2013). Men, in contrast, are more risk tolerant than women (Finucane et al. 2000). Although gender inequality prevails in different ways around the world, women's safety concerns for their family have been well documented in both environmental protection movements and neighborhood emergency preparedness campaigns (Litt et al. 2012; Luft 2008; Erikson 1994; Turner et al. 1986). Therefore, it is necessary to provide more diverse options for house retrofitting for families to increase their potential willingness to improve the anti-seismic resilience of their houses."

15. Throughout the figures need more context and linking to the text (e.g., Figure 8 on line 200 – it's hard to link the text to the figure. The figures need more explanatory captions to guide the reader, and the text needs more explanatory linking to the figures.

Ans: Thank you for the comment. Figures 8 to 10 could only provide limited information, and therefore, the revised version has deleted the figures. In addition, the whole section of the discussion was rewritten to improve the overall research purpose.

16. Other issues (such as fatalism or anxiety) need to be raised in the discussion in more detail – see e.g., McClure et al 2001 and Paton 2005, Wei & Lindell 2017

Ans: Thank you for the comment. The revised version focuses on discussing the potential pre- and postdisaster impacts of socioeconomic characteristics. Therefore, the research question is in the introduction, and other issues are addressed.

17. Line 227 – I'm not sure where the concept of bounded rationality came from in this paragraph. Needs better linking and explanation.

Ans: Thank you for the comment. The revised version focuses on discussing the potential pre- and postdisaster impacts of socioeconomic characteristics. Therefore, the research question is stated in the introduction, and bounded rationality is addressed.

18. Line 229 – what do you mean by internal control?

Ans: Thank you for the comment. The revised version focuses on discussing the potential pre- and postdisaster impacts of socioeconomic characteristics. Therefore, the research question is stated in the introduction, and internal control is addressed.

19. Line 228-231 repeats exactly some sentences in the introduction – reword appropriately

Ans: Thank you for the comment. Both section 3 and section 4 have been rewritten in the revised version.

20. The discussion needs some more explanation of the limitations – a limitation and future research section would be ideal. They are touched on in the conclusion (e.g., time limitation), but lack enough detail for the reader to evaluate and interpret.

Ans: Thank you for the comment. The revised version has added the explanations of the limitation in the conclusion section (please see lines 262-265).

"This study has the following limitations: the results might not be applicable to any other disaster events, only earthquakes. In addition, due to time limitations, the interviewees in the pre- and postearthquake surveys were different. There are potential topics that could be extended in future studies, such as the correlation between socioeconomic characteristics and the causes and effects of risk perceptions on adaptation behaviors."

21. The authors introduce the term 'subjective resilience' but need to define and explain this further. How does it relate to the various measures discussed?

Ans: Thank you for the comment. "Subjective resilience" has been discussed in the introduction, but it has less connection with the rest of the discussion. Therefore, the revised version has deleted such discussion in section 1 and focused on risk perceptions and adaptive behaviors (please see lines 35-59).

"It is necessary to minimize disaster risk and build resilience by self-evaluating the capabilities and capacities in responding to risk, that is, preparedness (Jones and Tanner 2017). Being prepared for a future disaster requires various components, such as sufficient personal character, social connections, and financial affordability (Baker and Cormier, 2015). People who are included in vulnerable minority groups and marginalized people might not be able to prepare in advance (Blake et al., 2017). Therefore, an increasing number of studies have emphasized measuring risk perceptions at the individual and household levels (Brown and Westaway 2011; Adger et al. 2009). The perception of disaster risk does not represent a direct function of the probability that threatening events will occur; rather, risk perception captures many other factors, such as attitude, cognition, the degree of danger comprehension, and vulnerability (Sjöberg 2000; Sjöberg 1996; Eagly and Chaiken 1993). Despite the substantial literature illustrating the origin (Barrows, 1923), concept (Sjöberg 2000; Sjöberg 1996), formation (Lindell et al., 2016; Whitney et al., 2004; Wu and Lindell, 2004; Lindell and Perry, 2000), and physical and social contexts of disaster risk perceptions (Blanchard-Boehm and Cook, 2004; Peacock et al., 2005; Peacock, 2003), less attention has been paid to systematically examining changes in risk perceptions.

In fact, disaster experiences might facilitate or constrain preparedness (Becker et al., 2017; Ejeta et

al., 2015; Lindell and Perry, 2011; Bostrom, 2008), and such effects might be biased across disasters, cultures or regions. A disaster resulting in limited impacts or the assumption that a future disaster will not occur might encourage people to not prepare for future disasters (Paton et al., 2014; Barron and Leider, 2010). Alternatively, people might take any adaptation approaches based upon damage or losses, physical injury, emotional injury and so on (Perry and Lindell, 2008; Nguyen et al., 2006; Heller et al., 2005). The physical damage or losses (Solberg et al., 2010) and psychological fear or anxiety (Rüstemli and Karanci, 1999) resulting from disaster experiences could motivate adaptation behaviors. However, socioeconomic characteristics such as income, age, and gender might encourage or discourage individuals from taking adaptive actions (Bankoff 2006; Wisner et al. 2004). For example, if people cannot act adequately to mitigate such anxiety, they might take no actions at all (Paton and McClure, 2013). Due to limited knowledge and resources, people tend not to respond to common disasters and tend to have personal preferences for disasters, such as denying disasters, denying disaster probability, and having certain beliefs about the government and public infrastructure. Therefore, examining risk perceptions and adaptation behaviors based on various socioeconomic characteristics could provide important information for disaster management."

22. The authors refer to 'sex' (see section 3.1 in particular), when I believe they need to be referring to 'gender' as the issues here relate to familial responsibilities and social roles relating to someone's gender – not their biological sex. See Rushton et al. (2019) for more.

Ans: Thank you for the comment. According to previous articles, both sex and gender could be applied for this category. However, gender indeed might be more appropriate, and the appropriate revisions have been made.

23. Figures need improving for clarity. Figure 1a-d need to be larger as the keys are hard to read, Figure 2 would be better in an appendix. All figures would benefit from more explanatory extensive captions that enable them to be read and interpreted more easily. Figures 3, 4, 5, 6 & 7 are not in a format I am familiar with. It took a while to interpret them, I feel they need some much clearer captions and further explanation to facilitate interpretation.

Ans: Thank you for the comment. We strengthened Figure 1a, and we removed Figures 1b-d based on the other reviewers' comments because the figures could only reveal limited information. Figures 3 to 7 are new here, and we took the comments seriously and improved the tables in the revised version (please see Figure 1 and Table 3 to Table 10)

24. Some of the tables are a bit unclear, e.g., Table 2: it is hard to follow which of the rows in column 1 apply to which rows in the other columns. Can they be

reformatted to aid comprehension?

Ans: Thank you for the comment. The revised version has improved the table to increase readability (please see Table 2)

"**Table 2** Sample characteristics in the pre- and postearthquake surveys.

| Characteristics | Pre- | Post- | Study area | Characteristics | Pre- | Post- | Study area |
|---|---|---|---|---|---|---|---|
| **Gender** | | | | **Occupation*** | | | |
| Male | 53.38% | 44.89% | 49.27% | Students | 9.09% | 7.23% | 38.53% |
| Female | 46.42% | 55.11% | 50.73% | Home Makers | 10.96% | 18.94% | |
| **Age** | | | | White-collar Workers | 37.76% | 32.55% | 59.08% |
| < 15 yr. | 7.46% | 1.70% | 13.97% | Blue-collar Workers | 41.96% | 41.28% | |
| 15-40 yr. | 38.23% | 28.30% | 37.96% | **House Ownership*** | | | |
| 40-60 yr. | 37.53% | 51.91% | 32.16% | Self-owned | 48.95% | 63.62% | 85.93% |
| > 60 yr. | 16.78% | 18.09% | 15.91% | Family-owned | 32.17% | 32.34% | 3.20% |
| **Education** | | | | Renting | 18.65% | 4.04% | 7.82% |
| Elementary/Junior High | 21.68% | 21.91% | 21.63% | | | | |
| High School | 47.32% | 41.49% | 30.54% | | | | |
| University/Graduate | 31.00% | 36.60% | 46.96% | | | | |

Note 1: The values without official statistics are replaced by data from the Tainan Municipality.
Note 2: The share of illiterate individuals in the study area is 0.87%.
Note 3: The official statistics for occupation are categorized into employment and unemployment, and the unemployment percentage is 2.39%. In addition, neither students nor home makers are included in the labor force.
Note 4: The official statistics for house ownership include self-owned, family-owned, renting, and others, and the percentages are 85.93%, 3.20%, 7.82%, and 3.05%, respectively.."

25. References ========

Rushton A, Gray L, Canty J, & Blanchard K (2019) Beyond binary: (re)defining "gender" for 21st century disaster risk reduction research, policy, and practice Int J Environ Res Public Health, https://doi.org/10.3390/ijerph16203984.

McClure J, Allen MW, & Walkey F (2001) Countering fatalism: Causal information in news reports affects judgments about earthquake damage Basic Appl Soc Psych, https://doi.org/10.1207/S15324834BASP2302_3.

Paton D, Smith L, & Johnston D (2005) When good intentions turn badâ˘A´r: promoting natural hazard preparedness Aust J Emerg Manag 20(1) 25–30.

Wei H-L, & Lindell MK (2017) Washington households' expected responses to lahar threat from Mt. Rainier Int J Disaster Risk Reduct 22 77–94, https://doi.org/10.1016/J.IJDRR.2016.10.014.

Senkbeil JC, Scott DA, Guinazu-Walker P, & Rockman MS (2014) Ethnic and Racial Differences in Tornado Hazard Perception, Preparedness, and Shelter Lead Time in Tuscaloosa Prof Geogr 66(4) 610–620, https://doi.org/10.1080/00330124.2013.826562.

Blake D, Marlowe J, & Johnston D (2017) Get prepared: Discourse for the privileged? Int J Disaster Risk Reduct 25 283–288, https://doi.org/10.1016/J.IJDRR.2017.09.012.

Olofsson A, & Rashid S (2011) The White (Male) Effect and Risk Perception: Can Equality Make a Difference? Risk Anal 31(6) 1016–1032, https://doi.org/10.1111/j.1539-6924.2010.01566.x.

Lindell MK, & Prater C (2002) Risk area resident' perceptions and adoption of seismic hazard adjustments J Appl Soc Psychol 32(11) 2377–2392.

Palmer CGS (2003) Risk perception: Another look at the "white male" effect Heal Risk Soc 5(1) 71–83, https://doi.org/10.1080/1369857031000066014.

Dooley D, Catalano R, Mishra S, & Serxner S (1992) Earthquake Preparedness: Predictors in a Community Survey J Appl Soc Psychol 22(6) 451–470, https://doi.org/10.1111/j.1559-1816.1992.tb00984.x.

Bateman JM, & Edwards B (2002) Gender and Evacuation: A Closer Look at Why Women Are More Likely to Evacuate for Hurricanes Nat Hazards Rev 3(3) 107–117, https://doi.org/10.1061/(ASCE)1527-6988(2002)3:3(107).

Lindell MK, & Perry RW (2012) The Protective Action Decision Model: Theoretical Modifications and Additional Evidence Risk Anal 32(4) 616–632, https://doi.org/10.1111/j.1539-6924.2011.01647.x.

Becker JS, Paton D, & Johnston DM (2014) Societal Influences on Earthquake Information Meaning-Making and Household Preparedness Int J Mass Emerg Disasters 32(2) 317–352.

Lindell MK, & Hwang SN (2008) Households ' Perceived Personal Risk and Responses in a Multihazard Environment , https://doi.org/10.1111/j.1539-6924.2008.01032.x.

Lindell MK, & Prater C (2002) Risk area resident' perceptions and adoption of seismic hazard adjustments J Appl Soc Psychol 32(11) 2377–2392.

Becker JS, Paton D, Johnston DM, Ronan KR, & McClure J (2017) The role of prior experience in informing and motivating earthquake preparedness Int J Disaster Risk Reduct 22 179–193, https://doi.org/10.1016/J.IJDRR.2017.03.006.

McClure J, Henrich L, Johnston DM, & Doyle EEH (2016) Are two earthquakes

better than one? How earthquakes in two different regions affect risk judgments and preparation in three locations Int J Disaster Risk Reduct 16 192–199, https://doi.org/10.1016/j.ijdrr.2016.03.003.

Paton D, Anderson E, Becker J, & Petersen J (2015) Developing a comprehensive model of hazard preparedness: Lessons from the Christchurch earthquake Int J Disaster Risk Reduct 14 37–45, https://doi.org/10.1016/j.ijdrr.2014.11.011.

Doyle EEH, McClure J, Potter SH, Becker JS, Johnston DM, Lindell MK, Johal S, Fraser SA, & Coomer MA (2018) Motivations to prepare after the 2013 Cook Strait Earthquake, N.Z. Int J Disaster Risk Reduct 31 637–649, https://doi.org/10.1016/j.ijdrr.2018.07.008.

Ans: Thanks for the long list for the references. We will include as many as we can according to our research question.

---

## Author Response (AR2)

Response to Editor's comments

1. Technical corrections are needed to clarify some aspects of the findings. I assume that where you have "p=0.000", this is because SPSS displays 0.000, but if you click on it, it will give you a precise number. Please go through the MS and tables, and change 0.000 to "<0.001". Simply 0.000 is not accurate.

Ans: Thanks for the recommendations. We have revisited SPSS and click on the P value and the P values for "fear of earthquake" in Gender and Occupation are 0.000000. Therefore, we have added the notes in the end of the tables and revised the values in the results. (Please see line 170, Table 3, and Table 7).

2. Please also go through the results section again and check for clarity - in particular, in the House ownership section, you state that people were more willing to retrofit after the EQ, which contradicts everything else! Please make sure that this section is very clear and reflects the tables and figures.

Ans: Thanks for the comments, and the description in house ownership is inaccurate. The results have been reviewed carefully to avoid any contradicts. (Please see line 226-228)

*"Overall, regardless of house ownership category, people tended to become more aware of earthquakes and less willing to retrofit their houses in the postearthquake survey."*

3. Finally, please revisit the conclusion and ensure that it is a clear summary of the findings. The "limitations" should be a new paragraph - at the moment it is too abrupt.

Ans: Thanks for the comments. Couple things in the conclusion section have not been revised last time. The section has been rewritten and added up a paragraph for limitation. (Please see line 273-287)

"      This study tends to explore the changes in risk perceptions and adaptation behaviors based on various socioeconomic characteristics before and after earthquake disasters. However, there are multiple limitations faced in this study. There are two surveys (October to December 2014, and May 2016) conducted in the study area. The predisaster survey was a street survey, while the postdisaster survey was a telephone survey based on phone number databases within the study area. Although the questions were the same in the two surveys, the interviewees in the pre- and postearthquake surveys were different. In addition, Meinong earthquake was a magnitude 6.6 earthquake caused 744 buildings reported as having been damaged, and in particular, one building fully collapsed, resulting in 115 deaths. It was a devastating earthquake but only caused one building fully collapsed. Such disaster experience might not necessary increase the awareness of buildings anti-seismic effect. The results

might not be applicable to any other disaster events, only earthquakes.

To sum up, the results can provide a general tendency regarding changes in risk perceptions and adaptation behaviors pre- and postdisaster events and the variations between different socioeconomic characteristics based upon Taiwanese disaster experience. The findings can serve as a reference to formulate risk communication strategies and for governments to make decisions on the prioritization of risk management policies. However, there are potential topics that could be extended in future studies, such as the correlation between socioeconomic characteristics, the causes and effects of risk perceptions on adaptation behaviors, and others."